# Murine endothelial serine palmitoyltransferase 1 (SPTLC1) is required for vascular development and systemic sphingolipid homeostasis

Andrew Kuo[1], Antonio Checa[2], Colin Niaudet[1], Bongnam Jung[1], Zhongjie Fu[3], Craig E Wheelock[2,4,5], Sasha A Singh[6], Masanori Aikawa[6], Lois E Smith[3], Richard L Proia[7], Timothy Hla[1]*

[1]Vascular Biology Program, Boston Children's Hospital, Department of Surgery, Harvard Medical School, Boston, United States; [2]Unit of Integrative Metabolomics, Institute of Environmental Medicine, Karolinska Institute, Stockholm, Sweden; [3]Department of Ophthalmology, Boston Children's Hospital, Harvard Medical School, Boston, United States; [4]Department of Respiratory Medicine and Allergy, Karolinska University Hospital, Stockholm, Sweden; [5]Gunma University Initiative for Advanced Research, Gunma University, Maebashi, Japan; [6]Center for Interdisciplinary Cardiovascular Sciences, Cardiovascular Medicine, Brigham and Women's Hospital, Harvard Medical School, Boston, United States; [7]Genetics and Biochemistry Branch, National Institute of Diabetes and Digestive and Kidney Diseases, National Institutes of Health, Bethesda, United States

*For correspondence: timothy.hla@childrens.harvard. edu

**Abstract** Serine palmitoyl transferase (SPT), the rate-limiting enzyme in the de novo synthesis of sphingolipids (SL), is needed for embryonic development, physiological homeostasis, and response to stress. The functions of de novo SL synthesis in vascular endothelial cells (EC), which line the entire circulatory system, are not well understood. Here, we show that the de novo SL synthesis in EC not only regulates vascular development but also maintains circulatory and peripheral organ SL levels. Mice with an endothelial-specific gene knockout of SPTLC1 (*Sptlc1* ECKO), an essential subunit of the SPT complex, exhibited reduced EC proliferation and tip/stalk cell differentiation, resulting in delayed retinal vascular development. In addition, *Sptlc1* ECKO mice had reduced retinal neovascularization in the oxygen-induced retinopathy model. Mechanistic studies suggest that EC SL produced from the de novo pathway are needed for lipid raft formation and efficient VEGF signaling. Post-natal deletion of the EC *Sptlc1* also showed rapid reduction of several SL metabolites in plasma, red blood cells, and peripheral organs (lung and liver) but not in the retina, part of the central nervous system (CNS). In the liver, EC de novo SL synthesis was important for acetaminophen-induced rapid ceramide elevation and hepatotoxicity. These results suggest that EC-derived SL metabolites are in constant flux between the vasculature, circulatory elements, and parenchymal cells of non-CNS organs. Taken together, our data point to the central role of the endothelial SL biosynthesis in maintaining vascular development, neovascular proliferation, non-CNS tissue metabolic homeostasis, and hepatocyte response to stress.

## Editor's evaluation

This study clearly defines an important role for SPTLC1 in vascular endothelial cell development. The data presented herein will help promote our understanding of endothelial cell metabolism and how metabolic disorders can cause vascular abnormalities.

## Introduction

Sphingolipids (SL), which comprise ~10–20% of total cellular lipids, are involved in wide range of biological processes. Complex membrane SL, sphingomyelin (SM) and glycosphingolipids, are enriched in lipid rafts and caveolae, which provide an optimal platform for intracellular signaling and cell-cell interactions (*van Meer et al., 2008*). In contrast, the secreted SL, sphingosine-1-phosphate (S1P), acts as a ligand for G-protein coupled receptors (S1PR1-5) which regulate many physiological and pathological functions (*Proia and Hla, 2015*). While SL can be derived from dietary sources, most cellular SL are synthesized via the de novo pathway. This pathway is initiated in the endoplasmic reticulum by the condensation of serine and fatty acyl-CoA to generate 3-ketosphinganine by the serine palmitoyltransferase (SPT) enzyme complex. The product, 3-ketosphinganine, is further converted into dihydrosphingosine (dhSph), dihydroceramide (dhCer), and ultimately ceramide (Cer), which is a substrate for the production of complex SL and metabolites such as sphingosine (Sph) and S1P (*Merrill, 2002*). The SPT complex, which is composed of long chain subunit 1 (SPTLC1), subunit 2/3 (SPTLC2/3) and small subunit A/B (SPTSSA/B), catalyzes the rate-limiting step in the de novo SL biosynthetic pathway (*Han et al., 2009*). Absence of any of the subunits abolishes the enzymatic activity of SPT, which is required for normal embryonic development in mice (*Hojjati et al., 2005*). In addition, SPT enzyme activity is controlled by interaction with ORMDL proteins, which respond to extracellular and cell-intrinsic cues to regulate the flux of SL synthesis by the de novo pathway (*Siow et al., 2015*; *Han et al., 2019*).

The involvement of de novo SL biosynthetic pathway in organ function and diseases have been deduced from genome wide association studies and mouse genetic models. Mutations in the *SPTLC1* gene are associated with peripheral neuronal dysfunction including hereditary sensory neuropathy type 1 (HSAN1) (*Bejaoui et al., 2001*; *Dawkins et al., 2001*), childhood amyotrophic lateral sclerosis (ALS) (*Mohassel et al., 2021*) and macular telengectasia type-2 (*Gantner et al., 2019*). *Sptlc2* haploinsufficiency in macrophages causes reduction of circulating SM and enhancement of reverse cholesterol transport in murine genetic models (*Chakraborty et al., 2013*), suggesting the requirement of this pathway in maintaining systemic SL homeostasis. Loss of function of the SPT complex in hepatocytes reduces SM in the plasma membrane and suppresses the expression of E- and N-cadherin, adherens junction constituents (*Li et al., 2016*), highlighting the importance of this pathway in maintaining proper plasma membrane function. Collectively, these data reveal the plethora of cell-type-specific functions of SL derived from the de novo pathway.

Endothelial cells (EC) line the inner layer of the circulatory system and mediate vascular tone control, barrier integrity, oxygen supply, immune cell trafficking and waste removal. The vasculature of the central nervous system (CNS), which includes the brain and retina, is unique in that EC help form a highly selective barrier for circulatory metabolites. Indeed, EC in CNS forms a specialized structure between the blood and the tissue, known as the blood brain barrier (BBB) and blood retinal barrier (BRB) that consist of cellular compartments including EC, pericytes, and neural cells (astrocytes, glia and neurons). SL metabolites, originally isolated from brain due to their relative abundance, are critical for CNS development, physiology and disease (*Olsen and Færgeman, 2017*). Metabolism of some lipids such as cholesterol are compartmentalized in CNS and non-CNS tissues (*Orth and Bellosta, 2012*). Whether SL metabolism in CNS and non-CNS organs is compartmentalized and the role of SL in organotypic vascular beds is not known.

Our current knowledge about SL metabolism in EC has been centered around S1P, a key lipid mediator needed for developmental and pathological angiogenesis, and Cer, a mediator of stress responses (*Cartier and Hla, 2019*; *Jernigan et al., 2015*). Activation of S1P signaling in the EC S1PR1 suppresses vascular endothelial growth factor (VEGF) dependent vascular sprouting and maintains barrier integrity, resulting in stable and mature vessels (*Jung et al., 2012*; *Gaengel et al., 2012*). Upon irradiation, Cer aggregate to form Cer-enriched membrane domains within the cell membrane, which regulate physicochemical properties of membranes, leading to apoptosis and capillary leak (*Kolesnick and Fuks, 2003*). Moreover, VEGFR2 has been shown to colocalize with SL enriched lipid

rafts on the plasma membrane and impact VEGF-induced intracellular signal transduction processes (*Laurenzana et al., 2015*). These results suggest that multiple SL metabolites regulate vascular development (angiogenesis), homeostasis and diseases. Recently, it was that EC-specific deletion of *Sptlc2* results in lower blood pressure and reduced plasma Cer, implying the protective function of de novo SL biosynthetic pathway in vascular disease (*Cantalupo et al., 2020*). However, the role of de novo SL biosynthetic pathway in metabolic flux of SL metabolites in various tissue compartments and vascular development is not known.

To address these questions, we generated a mouse model in which de novo synthesis of SL can be abolished by tamoxifen-induced *Sptlc1* gene deletion in endothelial cells (*Sptlc1* ECKO). *Sptlc1* ECKO mice exhibited delayed retinal vascular development and reduced pathological angiogenesis in an oxygen-induced retinopathy model. This is due, at least in part, to defects in endothelial lipid rafts and impaired VEGF signaling. Moreover, *Sptlc1* ECKO mice showed rapid decline of many SL, including dhSph, dihydrosphingosine-1-phosphate (dhS1P) and Cer in plasma and red blood cells (RBC), demonstrating SL metabolic flux from EC to circulation. SL content of peripheral tissues such as lung and liver were also reduced upon *Sptlc1* EC deletion. Reduced Cer levels in *Sptlc1* ECKO mice protected against acetaminophen-induced hepatoxicity, suggesting the functional importance of EC SL supply in liver response to stress. Collectively, our data not only demonstrate the role of de novo synthesis of SL during vascular development, but also reveal a novel function of EC as a source to provide SL to the circulation and peripheral tissues for their proper functions and response to stress.

## Results

### Endothelial SPT enzyme supplies sphingolipids to vascular and non-vascular cells in the lung

We generated a mouse strain in which *Sptlc1* floxed mice are crossed with tamoxifen-inducible VE-cadherin Cre recombinase mice (*Sptlc1* ECKO, *Figure 1—figure supplement 1A*). Mice were administered tamoxifen at 6 weeks of age and followed for 10 weeks. We did not observe any changes in body weight or gross abnormalities (*Figure 1—figure supplement 1B*). Blood chemistry analyses indicated that *Sptlc1* ECKO mice have normal liver enzyme levels, kidney function and blood chemistry (*Figure 1—figure supplement 1C*). To determine the efficiency and specificity of *Sptlc1* gene deletion, we prepared EC (CD31[+]; CD45[-]) and non-EC (CD31[-]) from single-cell suspensions isolated from enzymatically digested lung tissues (*Figure 1A*). Flow cytometry analyses showed that isolated EC population was ~90% pure and non-EC population contains <10% of contaminating CD31[+] cells (*Figure 1B*). SPTLC1 expression was reduced markedly (~80%) in *Sptlc1* ECKO endothelium specifically (*Figure 1C–E*). In contrast, SPTLC1 expression in bone marrow which has low EC content did not show differences in WT vs. *Sptlc1* ECKO (*Figure 1—figure supplement 2*). These results demonstrate that SPTLC1 expression is almost completely suppressed in the vascular endothelium of *Sptlc1* ECKO mice.

We next measured SL metabolites in isolated lung EC from *Sptlc1* ECKO mice by LC-MS/MS. SL species in the de novo biosynthetic pathway, namely, dhSph, C16-dihydroceramide (dhC16-Cer) were markedly reduced (*Figure 1F*). In addition, Sph, S1P and dhS1P were reduced. Cer species, including C16:0, C22:1, C22:0, C24:0, C24:1, and C26:0, were also reduced, resulting in decreased total EC Cer levels (*Figure 1G*, *Figure 1—figure supplement 3A*). Moreover, SM species, including C16:0, C22:0, C24:0, C26:1, C26:0 SM, were reduced, resulting in decreased total EC SM content (*Figure 1H*, *Figure 1—figure supplement 3B*). These data show that lack of SPTLC1 subunit leads to marked reduction in many SL metabolites of EC.

We noticed that dhSph, S1P and several Cer species were significantly reduced in non-EC populations isolated from lungs of *Sptlc1* ECKO mice while some SL (dhC16-Cer, dhS1P, Sph and SM) were unaffected (*Figure 1I–K*, *Figure 1—figure supplement 3C, D*). Given that non-EC population contained less than 10% of CD31[+] cells (*Figure 1B*), the reduction is unlikely to be caused by EC contamination. Instead, this finding suggests that EC-derived SL metabolites are transferred to non-EC parenchymal cells in the lung.

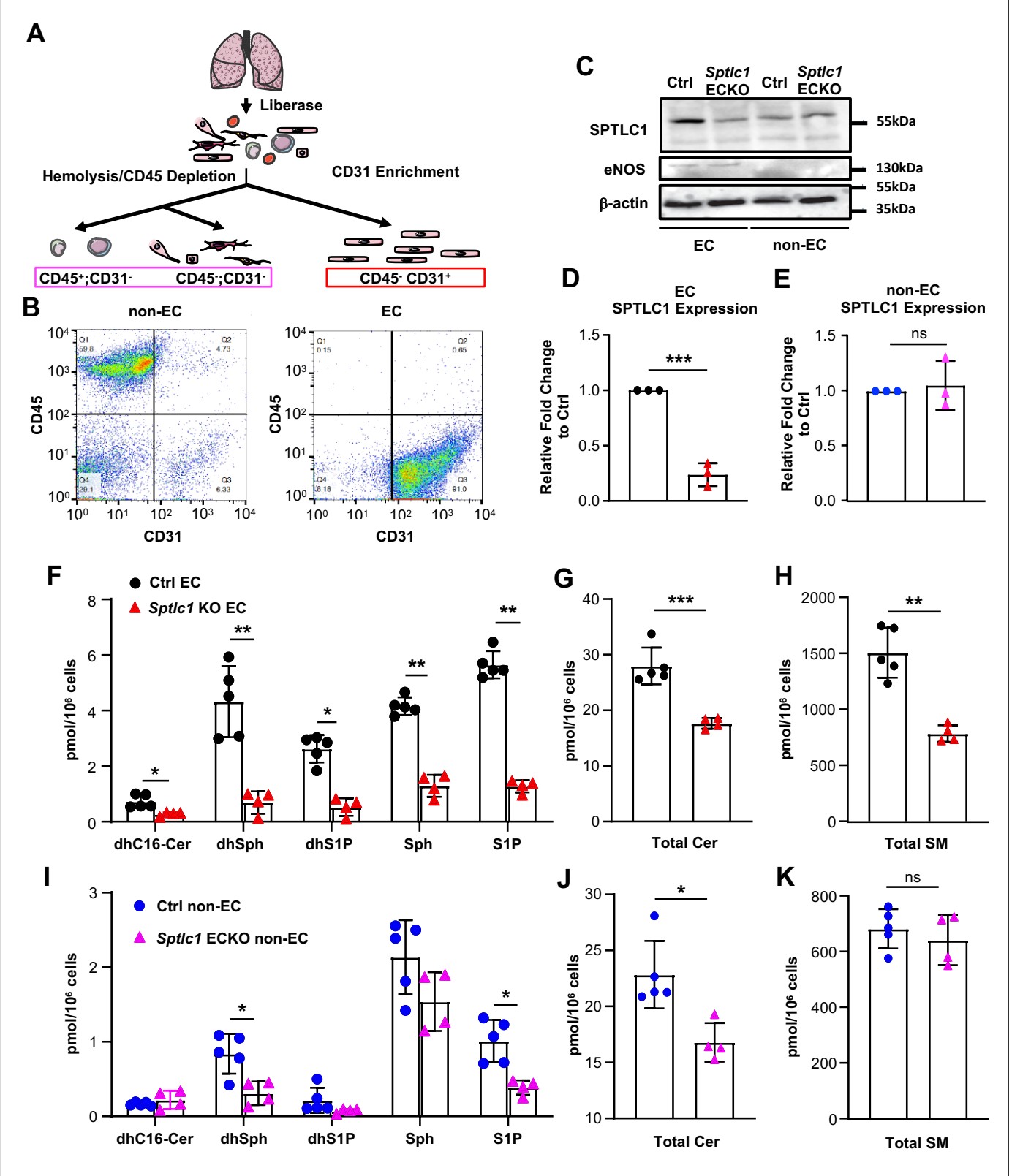

**Figure 1.** Ablation of *Sptlc1* in EC reduces SL content in both EC and non-EC cell population of the lung. (**A**) Scheme of lung EC and non-EC isolation procedure used in this figure. (**B**) Representative flow cytometry analysis for determination of lung non-EC and EC population purities. Non-EC is identified as CD45+/-; CD31- cells (Q1 + Q4). EC is identified as CD45-; CD31 + cells (**Q3**) (n=4 for both EC and non-EC). (**C**) EC and non-EC from Ctrl and *Sptlc1* ECKO lungs were analyzed by Western Blotting for SPTLC1 and eNOS expressions. eNOS was used as an EC marker to demonstrate purity

*Figure 1 continued on next page*

Figure 1 continued

of each population. β-actin was used as an internal control. Relative fold change of SPTLC1 to Ctrl in EC and non-EC population were normalized to β-actin and quantified in (**D**) an (**E**) (n=3 for both genotypes). SL content in Ctrl and *Sptlc1* KO EC was determined by LC-MS/MS, including dhC16-Cer, dhSph, dhS1P, Sph and S1P in (**F**), total Cer content in (**G**), and SM in (**H**) (n=5 for Ctrl EC, n=4 for *Sptlc1* KO EC). SL content in Ctrl and *Sptlc1* KO non-EC was determined by LC-MS/MS, including dhC16-Cer, dhSph, dhS1P, Sph, and S1P in (**I**), total Cer content in (**J**), and total SM content in (**K**) (n=5 for Ctrl non-EC, n=4 for *Sptlc1* KO non-EC). Values of lipidomic results can be accessed in *Figure 1—source data 1*. Data are expressed as mean ± SD. Statistical significance was determined by unpaired t test. *p<0.05; **p<0.01. ***p<0.001. ns, non-significant.

The online version of this article includes the following source data and figure supplement(s) for figure 1:

**Source data 1.** Levels of dhC16-Cer, dhSph, dhS1P, Sph, S1P, Cer, and SM in lung EC and non-EC populations of Ctrl and *Sptlc1* ECKO mice.

**Source data 2.** Original blots of EC and non-EC from Ctrl and *Sptlc1* ECKO lungs.

**Figure supplement 1.** *Sptlc1* ECKO mice exhibit similar body weight and blood chemistry as control mice.

**Figure supplement 2.** SPTLC1 expressions are not affected in bone marrow of *Sptlc1* ECKO mice.

**Figure supplement 2—source data 1.** Original blots of lysates from Ctrl and *Sptlc1* ECKO bone marrows.

**Figure supplement 3.** Cer and SM are reduced in *Sptlc1* KO EC while they are mildly affected in non-EC.

**Figure supplement 4.** eNOS expressions and phosphorylation of eNOS are not affected in *Sptlc1* KO EC.

**Figure supplement 4—source data 1.** Original blots of EC and non-EC from Ctrl and *Sptlc1* ECKO lungs.

## EC SPT regulates retinal vascular development and pathological neovascularization

We next investigated whether EC SPT is important for vascular development. Vascular network develops in the murine retina after birth and is fully formed by postnatal day 30 (P30) (*Rust et al., 2019*). *Sptlc1* ECKO mice exhibited a lower retinal vascular density and delayed radial expansion of retinal vascular network at P6 compared to controls (*Figure 2A–C*) while pericyte coverage was not altered (*Figure 2D and E*). Proliferation of retinal EC was reduced in *Sptlc1* ECKO mice as indicated by phosphorylated histone H3 staining (*Figure 2F and G*). Tip cells, which require active VEGF signaling during vascular development, was markedly reduced in *Sptlc1* ECKO mice, as indicated by ESM1 staining of developing retina (*Figure 2F and H*). Reduced tip cell formation and proliferation of stalk cells likely led to the abnormalities in retinal vascular development. The deeper vascular plexus, which begins to form ~P9 by vertical expansion of vascular sprouts from the superficial vascular network, also showed marked developmental delay in *Sptlc1* ECKO mice compared to controls (*Figure 2I–K*). However, expression of EC proteins important for BRB formation, including LEF1 (a key transcription factor involved in canonical Norrin/Wnt signaling), TFRC (transferrin receptor C), and CLDN5 (Claudin-5, a tight junction protein) were not altered at P6, suggesting that retinal EC organotypic specialization is not affected in the absence of SPT (*Figure 3A–E*). Interestingly, MFSD2A, a lysophospholipid transporter that suppresses transcytosis in EC, was upregulated in the retinal arteries of *Sptlc1* ECKO retina (*Figure 3D and F*). The delayed vascular developmental phenotype was normalized at P15 (*Figure 3—figure supplement 1*). We surmise that SL synthesized by other cells of the retina or derived from circulatory sources rescued the *Sptlc1* ECKO phenotype in the vasculature at later stages of development. Together, these results suggest that active SPT function, which supplies SL in the endothelium is needed for normal vascular development.

Oxygen-induced retinopathy (OIR) is a widely used murine disease model to investigate pathological neovascularization during retinopathy of prematurity, diabetic retinopathy and wet form of age-related macular degeneration (*Smith et al., 1994*; *Campochiaro, 2013*). VEGF produced by Müller glia and astrocytes is a major contributor to pathological vascular tuft formation in this model (*Weidemann et al., 2010*). We used this OIR model in WT and *Sptlc1* ECKO mice to determine the role of EC-derived SL metabolites in pathological angiogenesis. Tamoxifen was administered at P1-P3 to induce *Sptlc1* deletion and mouse pups were exposed to 75% $O_2$ from P7-P12 (*Figure 4A*). This hyperoxia environment causes capillary dropout (vaso-obliteration) due in part to suppression of VEGF production (*Smith et al., 1994*). In the absence of EC de novo SL synthesis, vascular dropout was exacerbated at P10 and P12, resulting in reduced vascular network density and increased avascular areas (*Figure 4B–G*). These results suggest that reduced SL de novo synthesis in EC led to reduced VEGF sensitivity of EC resulting in higher vascular dropout. Upon return to normoxia, avascular zones become severely hypoxic and induce high levels of VEGF which drive the formation of

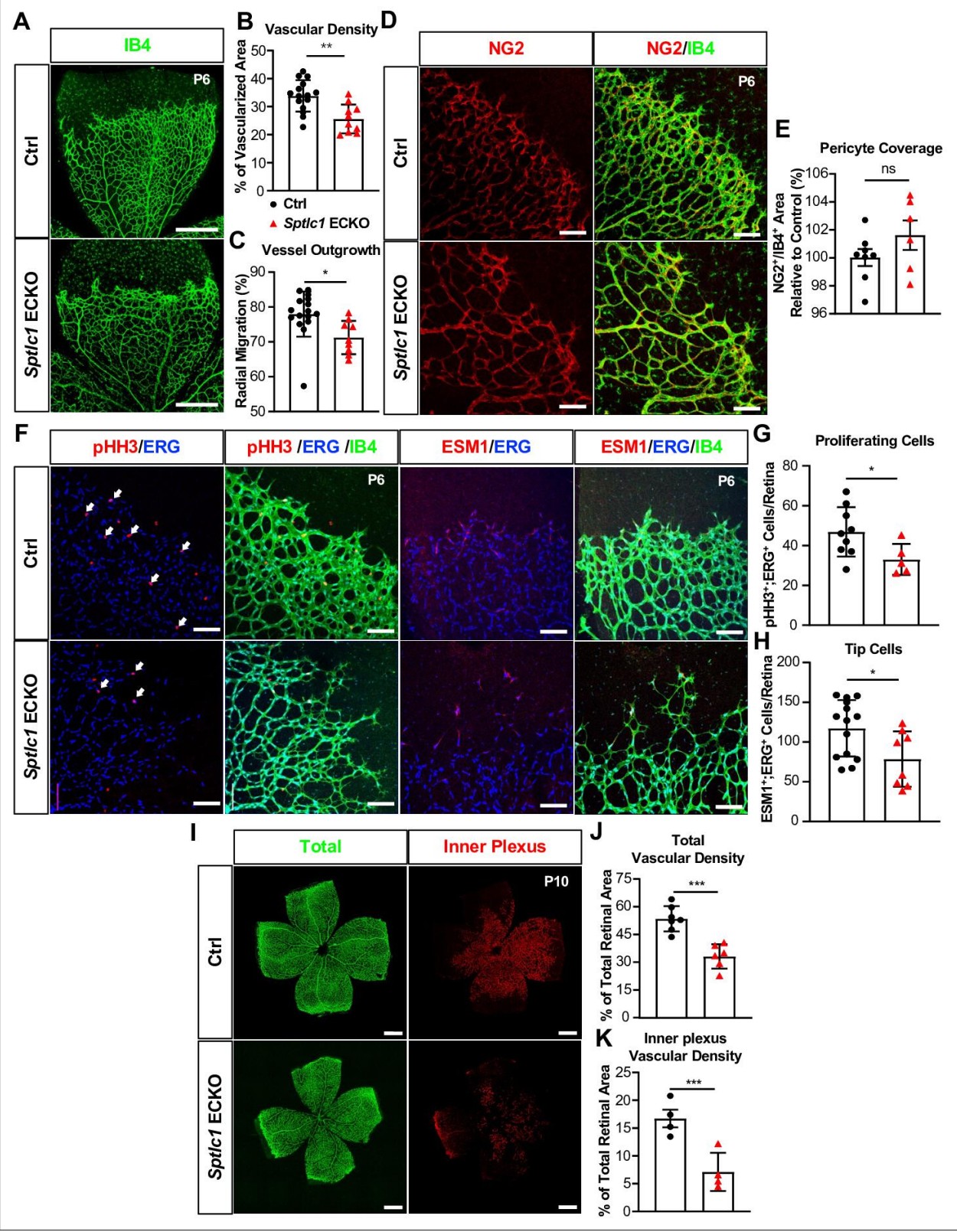

**Figure 2.** Retinal vascular development is delayed in the absence of *Sptlc1* in EC. (**A**) Retinal vascular plexus at P6 of Ctrl and *Sptlc1* ECKO mice were immunostained with Isolectin-B4 (IB4). Vascular density and outgrowth were quantified in (**B**) and (**C**) (n=16 for Ctrl, n=9 for *Sptlc1* ECKO). (**D**) Pericytes were immunostained with NG2 and pericytes coverage was quantified by NG2⁺/IB4⁺ area as shown in (**E**) (n=8 for Ctrl, n=6 for *Sptlc1* ECKO). (**F**) Proliferating cells and tip cells were immunostained with phospho-histone H3 (pHH3) and ESM1 respectively. ERG immunostaining was served as an

*Figure 2 continued on next page*

Figure 2 continued

EC marker. White arrows indicate pHH3$^+$/ERG$^+$ cells and quantified in (**G**) (n=8 for Ctrl, n=6 for *Sptlc1* ECKO). Tip cells were quantified by ESM1$^+$/ERG$^+$ cells as shown in (**H**) (n=14 for Ctrl, n=8 for *Sptlc1* ECKO). (**I**) Retinal vascular plexus at P10 of Ctrl and *Sptlc1* ECKO mice were immunostained with IB4. Representative images of total and inner plexus were shown and vascular density was quantified in (**J**) and (**K**) (n=4 for Ctrl, n=4 for *Sptlc1* ECKO). Data are expressed as mean ± SD. Statistical significance was determined by unpaired t test. *p<0.05; **p<0.01; ***p<0.001. ns, nonsignificant. Scale bar in (**A**): 500 μm, in (**D**) and (**F**): 100 μm, in (**I**) and (**L**): 500 μm.

vascular tufts. To examine the role of EC *Sptlc1* in the neovascular phase of OIR, we induced EC *Sptlc1* deletion at P12 which is the beginning of the retinal hypoxia in this animal model (*Figure 4H*). With this timed gene deletion strategy, we observed significant attenuation of pathological neovascularization in *Sptlc1* ECKO mice at P17 (*Figure 4I and J*). In addition, vaso-obliterated regions were larger in *Sptlc1* ECKO mice (*Figure 4I and K*). Collectively, the results from the OIR model indicate that de novo synthesis of SL in EC determine the pathological phenotypes in the retinal vasculature (vessel dropout and pathological vascular tuft formation), likely due to reduced VEGF responsiveness.

## EC SPT maintains lipid rafts and enables optimal VEGF signaling

To demonstrate that VEGF signaling is impaired in *Sptlc1* KO EC of the retina, exogenous VEGF-A was administered to P6 mouse pups via intraocular injection. In concurrence with published reports (*Eilken et al., 2017*; *Gerhardt et al., 2003*), VEGF-treated capillaries in the vascular front were dilated with increased EC numbers as depicted by ERG staining 4 hr post-injection (*Figure 5A–B*). These phenotypes were attenuated in *Sptlc1* ECKO retinas, suggesting that VEGF responsiveness in EC is attenuated in the absence of de novo SL synthesis (*Figure 5A–C*). Moreover, exogenous VEGF-A induced ESM1 expression in the plexus regions of the vasculature (ectopic tip cells) (*Rocha et al., 2014*). Supporting our hypothesis, the number of ESM1$^+$ ectopic tip cells was reduced in *Sptlc1* ECKO retinas compared to the Ctrl retinas (*Figure 5A–C*). Collectively, our results indicate that SPTLC1 in retinal endothelium contributes to proper VEGF responsiveness.

To further demonstrate that VEGF signaling and its downstream target activation are attenuated by loss of SPT activity, human umbilical vein endothelial cells (HUVEC) were treated with Myriocin, an inhibitor of SPT complex, followed by VEGF activation. In agreement with a previous report (*Cantalupo et al., 2020*), VEGF mediated ERK phosphorylation was attenuated by 72 hr of Myriocin pre-treatment (*Figure 5D*, lane 1–3). Importantly, 6 hr incubation of dhSph rescued ERK phosphorylation abrogated by Myriocin, suggesting that the de novo synthesis pathway of SL is required for efficient VEGF signaling (*Figure 5D*, lane 3–4). Supplementation of other SL including Sph and Cer (C18:0) can also restored VEGF-induced ERK phosphorylation from the inhibitory effects of Myriocin (*Figure 5D*, lane 3, 5, 6).

VEGFR is stabilized and modulated by lipid rafts, lipid microdomains enriched in sterols and SL (*Zabroski and Nugent, 2021*). Therefore, we tested the hypothesis that loss of SPT activity reduces SL which are needed for lipid raft formation and thereby dampen VEGF signaling. To test this, lipid rafts were detected by fluorescently-labelled Cholera Toxin B (CTXb), which specifically binds to the glycosphingolipid GM1. In HUVEC, Myriocin treatment to inhibit SPT and SL synthesis suppressed the CTXb signal at the plasma membrane (Vascular Endothelial Cadherin (VECad) positive area) (*Figure 5F* left panels, quantified in G). This suggests that continuous SPT activity is needed to form GM1, a lipid raft constituent in HUVEC. In Myriocin treated cells, exogenous supplementation with dhSph and Sph for 6 hr rescued lipid rafts in the plasma membrane (*Figure 5F* right panels, quantified in G). Taken together, our in vivo and in vitro mechanistic studies demonstrate that de novo synthesis of SL is necessary for efficient VEGF signaling via modulation of lipid raft formation.

## Endothelial SPT activity is critical to maintain circulatory SL metabolic homeostasis

Since the non-EC parenchymal cells isolated from the lungs of *Sptlc1* ECKO mice contain reduced levels of SL metabolites (*Figure 1I–K*), we hypothesized that EC-derived SL are transferred to non-EC parenchymal cells either directly (cell-cell transfer) or indirectly via the circulatory system. We first quantified SL metabolites in plasma isolated from adult *Sptlc1* ECKO mice. S1P and dhS1P levels were reduced ~60% in *Sptlc1* ECKO (*Figure 6A*). This reduction is more pronounced than that observed in sphingosine kinase-1 and –2 ECKO (*Gazit et al., 2016*) and S1P transporter (SPNS2) ECKO mice

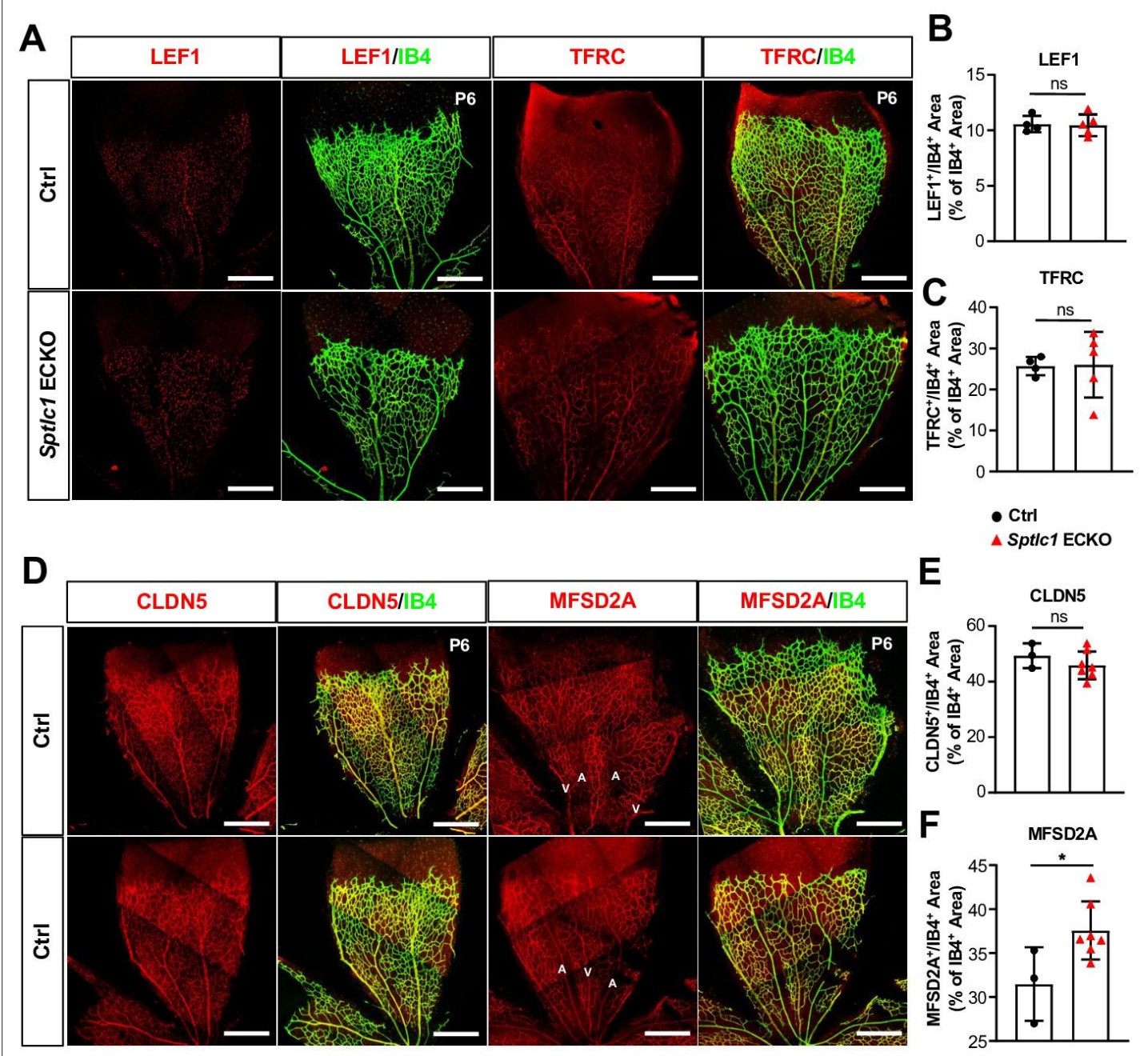

**Figure 3.** Blood-retina barrier (BRB) protein expressions are not reduced in the retina of *Sptlc1* ECKO mice. (**A**) BRB associated proteins including LEF1 and TFRC in Ctrl and *Sptlc1* ECKO P6 retina were immunostained and quantified by colocalization with IB4 staining shown in (**B**) and (**C**) (n=4 for Ctrl, n=5 for *Sptlc1* ECKO). (**D**) Tight junction protein, CLDN5 in Ctrl and *Sptlc1* ECKO P6 retina were immunostained and quantified by colocalization with IB4 staining shown in (**E**) (n=3 for Ctrl, n=7 for *Sptlc1* ECKO). BRB specific protein, MFSD2A, in Ctrl and *Sptlc1* ECKO P6 retina were immunostained and quantified by colocalization with IB4 staining shown in (**F**) (n=3 for Ctrl, n=7 for *Sptlc1* ECKO). Arterial and venous regions are marked as 'A' and 'V'. Data are expressed as mean ± SD. Statistical significance was determined by unpaired t test. *p<0.05; ns, non-significant. Scar bars, 100 μm.

The online version of this article includes the following figure supplement(s) for figure 3:

**Figure supplement 1.** Retinal vascular development is normalized at P15 in *Sptlc1* ECKO mice.

**Figure supplement 2.** Retinal vascular development is not affected in SK ECKO and *Apom-/-* mice.

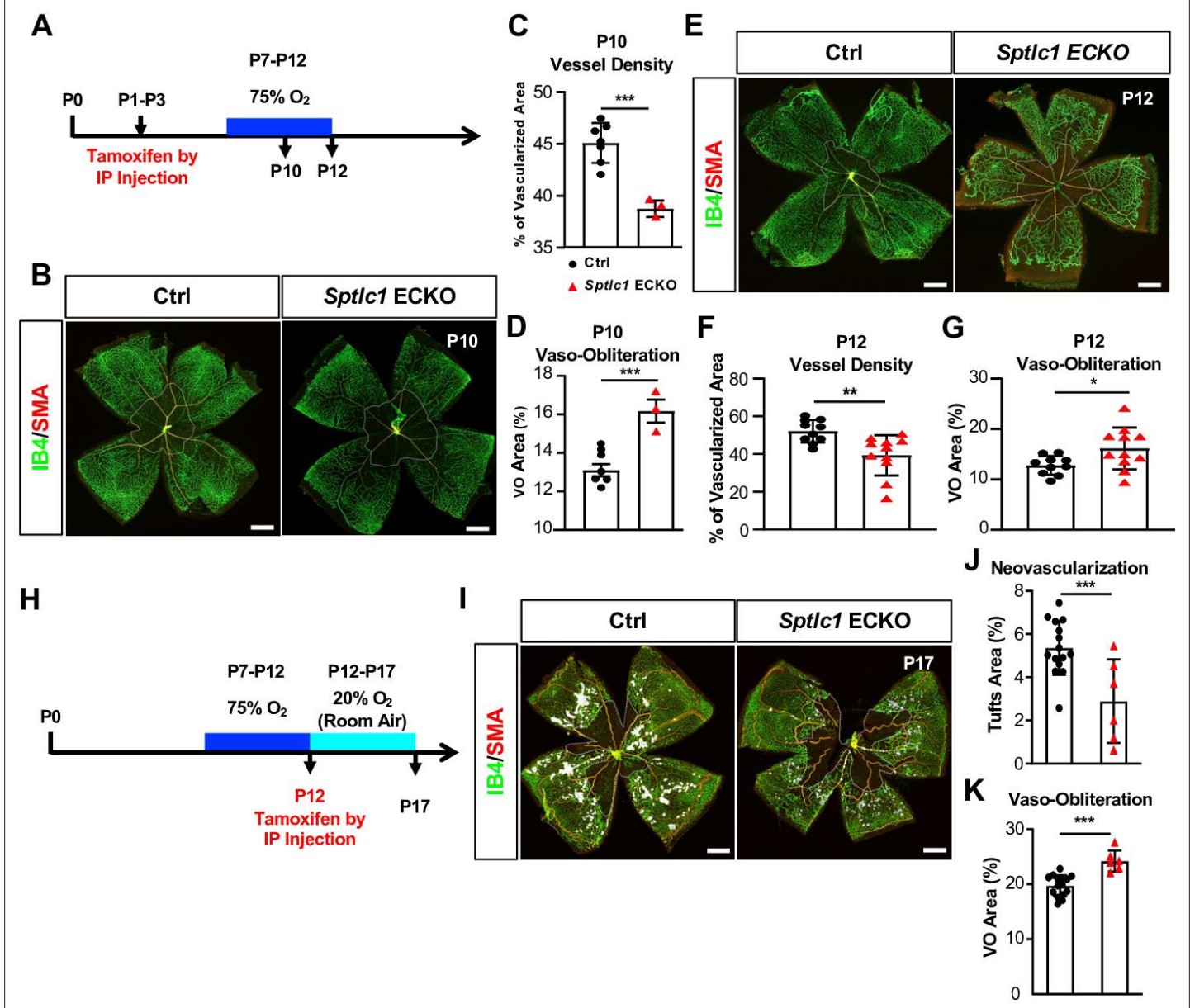

**Figure 4.** *Sptlc1* ECKO mice show enhanced vaso-obliteration and reduced neovascularization in oxygen induced retinopathy model. (**A**) Schematic of timeline for tamoxifen treatment, oxygen challenge and tissue collection used from (**B**) to (**G**). (**B**) P10 retinas of control and *Sptlc1* ECKO mice were immunostained with IB4 and smooth muscle actin (SMA). Vascular density and vaso-obliteration (dashed line area) were quantified in (**C**) and (**D**) (n=7 for Ctrl, n=3 for *Sptlc1* ECKO). (**E**) P12 retinas of control and *Sptlc1* ECKO mice were stained with IB4 and SMA. Vascular density and vaso-obliteration (dashed line area) were quantified in (**F**) and (**G**). (n=10 for Ctrl, n=11 for *Sptlc1* ECKO). (**H**) Schematic of timeline for tamoxifen treatment, oxygen challenge and tissue collection used from (**I**) to (**K**). Note that tamoxifen was administered after 75% O2 challenge. (**I**) P17 retinas of control and *Sptlc1* ECKO mice were immunostained with IB4 and SMA. Neovascularization (white area) and vaso-obliteration (dashed line area) were quantified in (**J**) and (**K**) (n=15 for Ctrl, n=6 for *Sptlc1* ECKO). Data are expressed as mean ± SD. Statistical significance was determined by unpaired t test. *p<0.05; **p<0.01, ***p<0.001. ns, non-significant. Scale bars, 500 µm.

(***Hisano et al., 2012***; ***Mendoza et al., 2012***). dhSph, a sphingoid base synthesized in the de novo synthesis pathway, was also reduced significantly (***Figure 6A***). Moreover, Cer species (C24:1, C24:0, C26:1, and C26:0) were reduced in *Sptlc1* ECKO plasma, which resulted in reduction of total Cer in plasma (***Figure 6B and C***). In contrast, SM species were not altered in the plasma of *Sptlc1* ECKO mice (***Figure 6D and E***). These results indicate that SL de novo synthesis pathway in EC is a significant source of sphingoid bases, their phosphorylated metabolites and Cer in plasma.

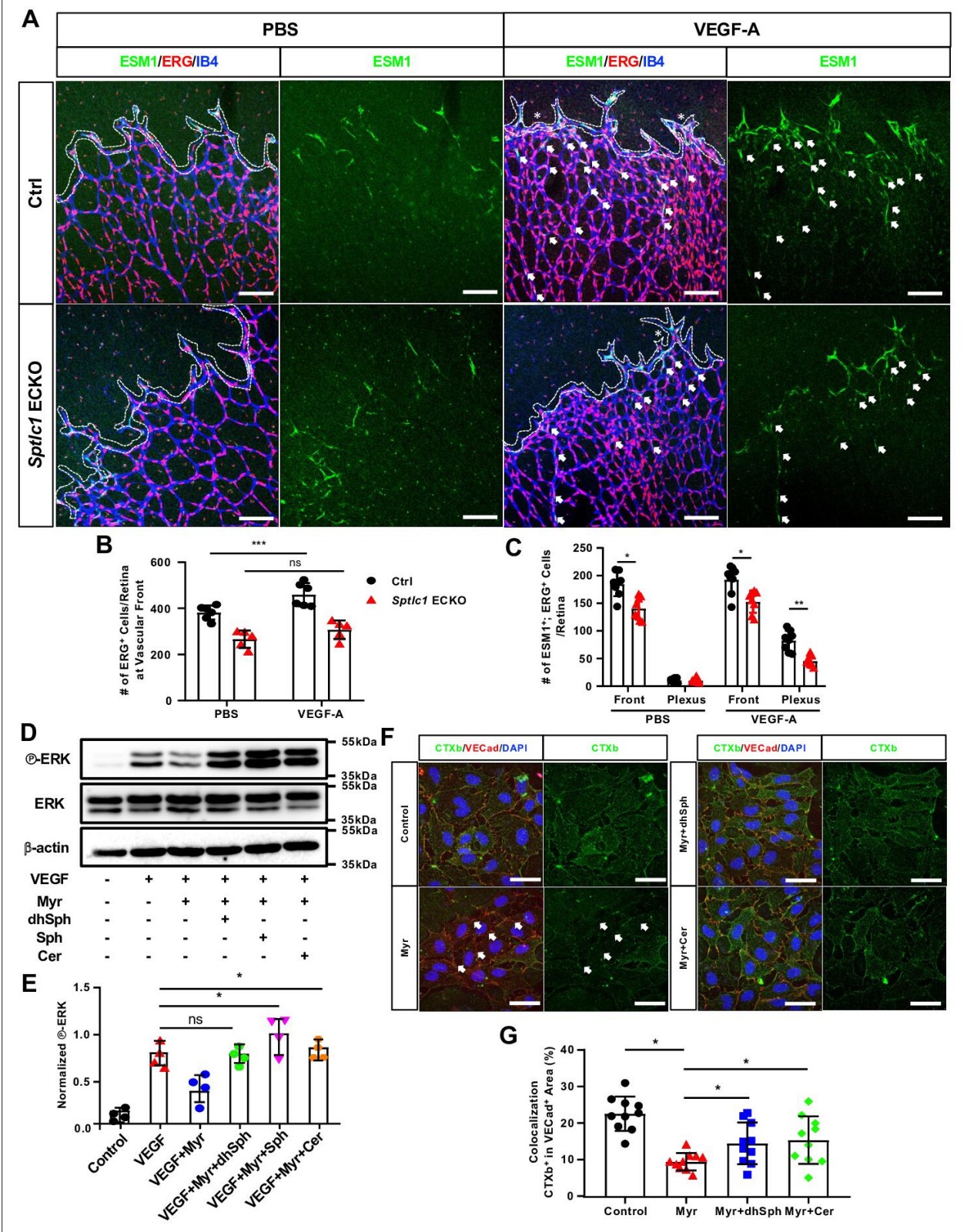

**Figure 5.** Loss of SPT activity in EC impairs VEGF signaling via modulation of lipid rafts. (**A**) P6 retinas of control and *Sptlc1* ECKO mice 4 hr post intraocular administration of mouse VEGF-A (50 ng/retina) or vehicle (PBS) were immunostained with ESM1, ERG and IB4. ERG numbers in the vascular front regions (dashed line area) were quantified in (**B**). Asterisks indicate dilated vessel area. ESM1+/ERG+ cells in the vascular front and capillary plexus (outside of dashed line area) were quantified in (**C**) (n=8 for Ctrl, n=8 for *Sptlc1* ECKO). Arrows indicate ectopic tip cells induced by VEGF treatment

*Figure 5 continued on next page*

*Figure 5 continued*

in capillary plexus regions. (**D**) HUVECs treated with Myriocin (Myr), dhSph, Sph and Cer followed by 5 min VEGF-A (50 ng/mL) were analyzed by western blotting for total and phospho-ERK. (**E**) Relative fold change to phosphorylation of ERK was first normalized to total ERK then toβ-actin as internal control (n=4). (**F**) HUVECs treated with Myr, dhSph, and Sph were immunostained with conjugated CTxb to label lipid rafts and VECad to label plasma membrane. Arrows indicate loss of CTXb signal in VECad$^+$ area by Myr treatment. Percentage of CTXb signal colocalization in VECad$^+$ area was quantified in (**G**) (n=10 in each condition from two independent experiment). Data are expressed as mean ± SD. Statistical significance was determined by unpaired t test. *p<0.05; **p<0.01; ***p<0.001, ns, non-significant. Scale bars in (**A**): 100 µm. and in (**F**): 50 µm.

The online version of this article includes the following source data for figure 5:

**Source data 1.** Original blots of HUVEC lysates treated with VEGF, Myr and different of SL.

RBC, which store and release S1P, are a major contributor of plasma S1P (*Pappu et al., 2007*). We therefore quantified SL metabolites in RBC of *Sptlc1* ECKO mice. Sphingoid bases and their phosphorylated counterparts (dhSph, dhS1P, and S1P) were significantly reduced in *Sptlc1* ECKO RBCs (*Figure 6F*). Cer species (C22:1, C22:0, C24:0, and C26:0) were also reduced (*Figure 6G and H*). SM species were not altered in RBC of *Sptlc1* ECKO mice (*Figure 6I and J*). These results show that reduction of SL metabolites in EC result in concomitant changes in plasma and RBC pools of SL metabolites. This suggests that SL metabolic flux occurs from EC to plasma and cells in circulation.

We next determined the kinetics of SL reduction in plasma/RBC after the deletion of EC *Sptlc1*. Tamoxifen was injected at 6 weeks of age to induce deletion of EC *Sptlc1* and plasma/RBC SL metabolites were quantified at various times. Rapid reduction of plasma S1P levels observed as early as 1 week and maintained thereafter (*Figure 7A*). Kinetics of RBC S1P reduction was similar (*Figure 7B*), suggesting that EC SL metabolites are rapidly secreted into plasma and that RBC and plasma SL pools are in rapid equilibrium under physiological conditions.

Plasma S1P is bound to two chaperones, namely Apolipoprotein M (ApoM) on HDL, and albumin in the lipoprotein free fraction (LPF) (*Murata et al., 2000*). ApoM and albumin levels were not altered in *Sptlc1* ECKO plasma (*Figure 7—figure supplement 1*). By fractionation of plasma using fast protein liquid chromatography, we determined that ApoM bound S1P was not affected while albumin bound S1P in the LPF was significantly reduced in *Sptlc1* ECKO mice (*Figure 7C*). Since albumin bound S1P is less stable (*Fleming et al., 2016*), released S1P from RBC or EC first bind to albumin, followed by stable interaction with ApoM. Nevertheless, these data suggest that EC SPT-derived S1P is secreted into plasma and equilibrates rapidly with RBC storage pools.

To determine whether the de novo synthesis of SL in EC also contributes to circulatory pools of SL species in early postnatal mice, we deleted EC *Sptlc1* from P1-P3 followed by measurement of plasma SL metabolites. Sphingoid bases and phosphorylated derivatives (dhSph, Sph, S1P, dhS1P) showed significant reduction from P14 to P28 in *Sptlc1* ECKO mice (*Figure 7D–G*). In contrast, plasma dhCer, Cer and SM were not changed (*Figure 7H–J*, *Figure 7—figure supplement 2*). Differential reduction in SL metabolites in plasma could be due to contributions from non-EC sources, such as the diet and metabolic flux from other cell types at different ages. Together, these results suggest that EC actively provide SL metabolites to circulation postnatally.

## SPT in EC supplies sphingolipids in liver and lung but not retina

We examined whether EC SPT enzyme loss affects SL metabolite levels in various organs. We analyzed the liver, lung, and retina because these organs are supplied by vessels with specialized organotypic endothelium (*Potente and Mäkinen, 2017*). The liver contains sinusoidal EC while lung vasculature is lined by EC specialized for gas exchange. In contrast, the retinal vasculature is lined by CNS-type endothelium which contains numerous tight junctions and specialized transporters with minimal transcytotic vesicles (*Andreone et al., 2017*). DhSph, a direct product of the de novo SL biosynthetic pathway was significantly reduced in lung and liver of *Sptlc1* ECKO mice (*Figure 8A and D*). In contrast, S1P and Sph levels in these tissues were not reduced (*Figure 8A and D*). Cer content was reduced about 30–50% in both organs from *Sptlc1* ECKO mice with C24:0 and C24:1 Cer species exhibiting largest changes (*Figure 8B, C, E and F*). These results suggest that EC supply SL metabolites from the de novo pathway into circulation which then equilibrates with liver and lung SL pools. We also measured SL content of retina, a CNS organ system. In sharp contrast to lung and liver, none of the SL metabolites were altered by lack of EC SPT (*Figure 8G–H*). These results reveal that the SL

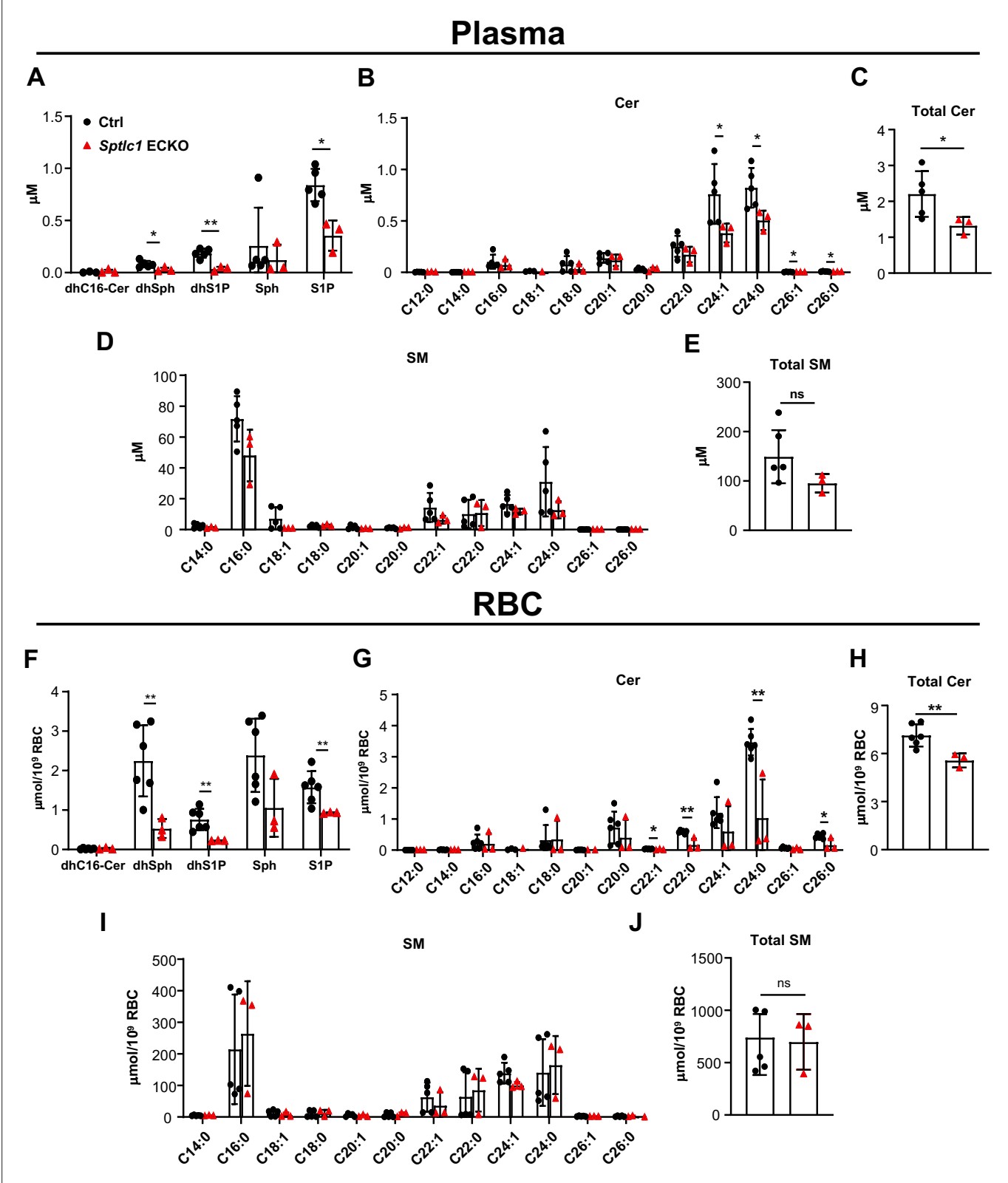

**Figure 6.** *Sptlc1* ECKO displays reduced SL content in circulation. SL content in plasma from Ctrl and *Sptlc1* ECKO mice was determined by LC-MS/MS, including dhC16-Cer, dhSph, dhS1p, Sph and S1P in (**A**), Cer with different fatty acyl chain length in (**B**), total Cer content in (**C**), and SM with different fatty acyl chain length in (**D**), and total SM in (**E**) (n=5 for Ctrl, n=3 for *Sptlc1* ECKO). SL content in RBC from Ctrl and *Sptlc1* ECKO mice was determined by LC-MS/MS, including dhC16-Cer, dhSph, dhS1p, Sph, and S1P in (**F**), Cer with different fatty acyl chain lengths in (**G**), total Cer content in (**H**), and SM

*Figure 6 continued on next page*

*Figure 6 continued*

with different fatty acyl chain lengths in (**I**), and total SM in (**J**) (n=5 for Ctrl, n=3 for *Sptlc1* ECKO). Values of lipidomic results can be accessed in *Figure 6—source data 1*. Data are expressed as mean ± SD. Statistical significance was determined by unpaired t test. *p<0.05; **p<0.01. ns, non-significant.

The online version of this article includes the following source data for figure 6:

**Source data 1.** Levels of dhC16-Cer, dhSph, dhS1P, Sph, S1P, Cer, and SM in plasma and RBC of Ctrl and *Sptlc1* ECKO mice.

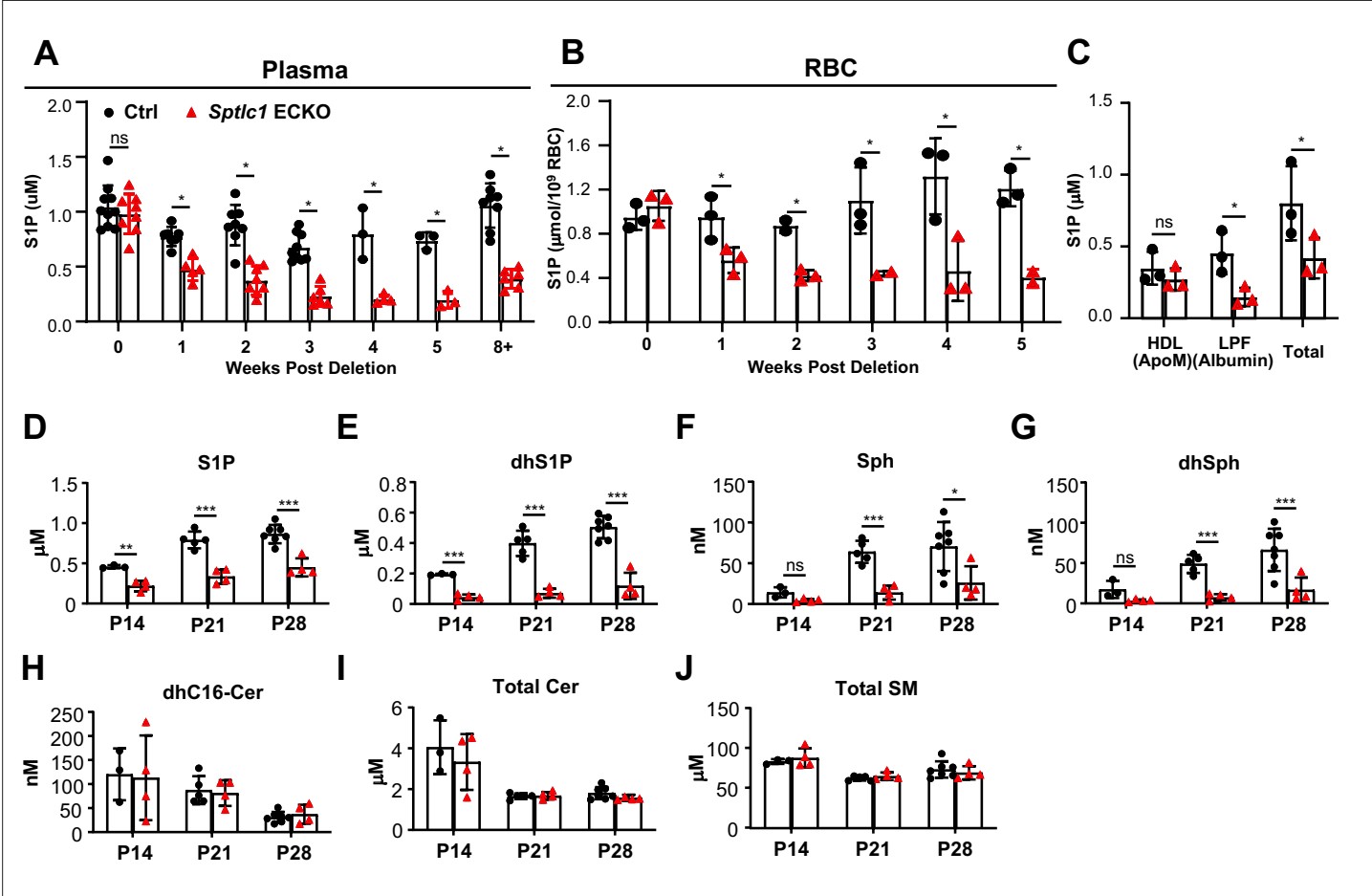

**Figure 7.** Rapid reduction of SL content in circulation upon *Sptlc1* deletion. S1P levels from plasma and RBC of Ctrl and *Sptlc1* ECKO mice were measured upon *Sptlc1* deletion over time as shown in (**A**) and (**B**) (n=3–8 for Ctrl in each time point, n=2–8 for *Sptlc1* ECKO in each time point). (**C**) Plasma from Ctrl and *Sptlc1* ECKO mice were fractionated by HPLC (Bio-Rad) into HDL (ApoM enriched), Lipoprotein Free (LPF) (Albumin enriched). S1P levels in each fraction, including total plasma were measured by LC-MS/MS (n=3 for Ctrl, n=3 for *Sptlc1* ECKO). SL content from plasma samples of Ctrl and postnatal (**P1–P3**) deletion of *Sptlc1* in EC was measured at P14, P21 and P28 by LC-MS/MS. Species measured are as followed: S1P (**D**), dhS1P (**E**), Sph (**F**), dhSph (**G**), dhCer (**H**), total Cer (**I**), and total SM (**J**) (n=3–7 for Ctrl in each age, n=4 for *Sptlc1* ECKO in each age). Data are expressed as mean ± SD. Statistical significances in (**A**) and (**B**) were determined by one-way ANOVA test. Values of lipidomic results can be accessed in *Figure 7—source data 1*. Statistical significances in (**C**)-(**G**) were determined by unpaired t test. *p<0.05; **p<0.01; ***p<0.001. ns, non-significant.

The online version of this article includes the following source data and figure supplement(s) for figure 7:

**Source data 1.** Levels of dhC16-Cer, dhSph, dhS1P, Sph, S1P, Cer, and SM in plasma and RBC of Ctrl and *Sptlc1* ECKO mice.

**Figure supplement 1.** Plasma FPLC fractionation indicates similar protein abundance of ApoM and Albumin in HDL and lipoprotein free fraction between Ctrl and *Sptlc1* ECKO mice.

**Figure supplement 1—source data 1.** Original blots from HDL and LPF fractions of plasma from Ctrl and *Sptlc1* ECKO mice.

**Figure supplement 2.** Plasma Cer and SM are not affected in postnatal deletion of *Sptlc1* ECKO mice.

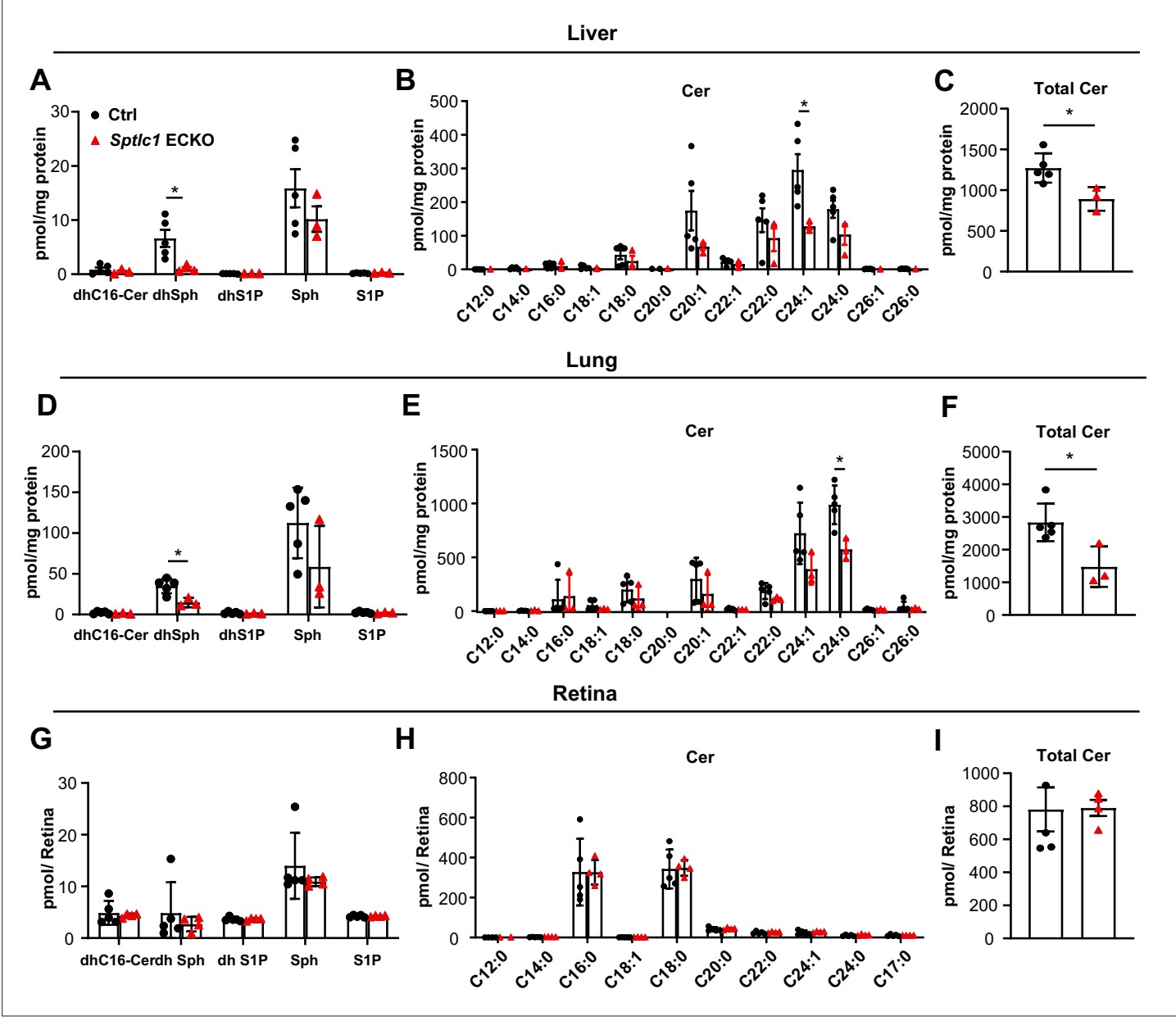

**Figure 8.** *Sptlc1* ECKO mice exhibit reduced SL content in peripheral but not CNS organs. SL content from adult liver homogenate of Ctrl and *Sptlc1* ECKO adult mice was measured by LC-MS/MS, including dhCer, dhSph, dhS1P, Sph and S1P (**A**), Cer with different fatty acyl chain length (**B**) and total Cer (**C**) (n=3 for Ctrl, n=3 for *Sptlc1* ECKO). SL content from adult lung homogenate of Ctrl and *Sptlc1* ECKO mice was measured by LC-MS/MS, including dhCer, dhSph, dhS1P, Sph, and S1P (**D**), Cer with different fatty acyl chain length (**E**) and total Cer (**F**) (n=3 for Ctrl, n=3 for *Sptlc1* ECKO). SL content from P28 retina homogenate of Ctrl and *Sptlc1* ECKO mice were measured by LC-MS/MS, including dhCer, dhSph, dhS1P, Sph, and S1P (**G**), Cer with different fatty acyl chain length (**H**) and total Cer (**I**) (n=5 for Ctrl, n=4 for *Sptlc1* ECKO). Values of lipidomic results can be accessed in *Figure 8— source data 1*. Data are expressed as mean ± SD. Statistical significance was determined by unpaired t test. *p<0.05.

The online version of this article includes the following source data for figure 8:

**Source data 1.** Levels of dhC16-Cer, dhSph, dhS1P, Sph, S1P, Cer, and from lung, liver and retina lysates of Ctrl and *Sptlc1* ECKO mice.

metabolic flux from EC to adjacent tissues only occurs in circulation and non-CNS organs such as the lung and the liver but not in the retina, which is part of the CNS.

## EC SPT supply of liver sphingolipids influences acetaminophen-induced liver injury

Liver function was normal in *Sptlc1* ECKO suggesting that reduced SL metabolites are not critical for hepatic homeostasis. However, chemical stress elevates SL metabolites such as Cer, which are thought to mediate liver injury (*Li et al., 2020*; *Park et al., 2013*). To determine if EC derived SL influence hepatocyte function, we examined hepatotoxicity induced by N-acetyl-para-aminophenol (APAP; a.k.a. Acetaminophen/Tylenol) in WT and *Sptlc1* ECKO mice. APAP treatment resulted in reduced liver injury (aspartate aminotransferase (AST) levels and liver necrosis) in *Sptlc1* ECKO mice compared with controls at early time points (4–8 hr) (*Figure 9A–C*). Liver SL metabolites were quantified at 8 and 24 hr after APAP administration. Many SL metabolites (dhCer C16:0, dhSph, dhS1P, Sph, S1P, Cer C24:1, and total Cer) were lower in the liver at 8 hr post-APAP but only dhCer C16:0 and dhSph were lower in the *Sptlc1* ECKO mice at 24 hr (*Figure 9D–I*). These data suggest that reduced Cer levels in the liver of *Sptlc1* ECKO mice at early stabges to attenuation of APAP-induced liver injury.

To determine the mechanisms involved in early protective response to APAP toxicity in *Sptlc1* ECKO mice, we examined levels of glutathione (GSH), the main antioxidant that protects from APAP-induced hepatoxicity. Liver GSH levels were elevated in *Sptlc1* ECKO mice prior to APAP injection. Even though GSH is diminished after APAP-administration, *Sptlc1* ECKO mice still had higher GSH levels than the WT counterparts at 8 hr (*Figure 9J*). Thus, reduced flux of SL metabolites from EC to liver results in attenuated drug-induced hepatotoxicity via modulation of antioxidant GSH levels. These data reveal that EC-derived SL impact stress response of the hepatocytes and liver injury, thus providing a functional significance of this metabolic pathway.

## Discussion

Lipids are derived both from the diet and from biosynthetic routes in organs. Their concentrations in various cells and tissues are regulated stringently to maintain normal physiological functions. The numerous SL species are no exception. Altered SL levels due to mutations in biosynthetic or degradative enzymes lead to embryonic lethality or severe genetic diseases (*Hannun and Obeid, 2018*). However, SL homeostatic mechanisms in various organ systems is poorly understood. For example, cell types which supply SL in various organ systems are not known. In this study, we show that SPT activity in EC regulates developmental and pathological angiogenesis. We also discover a novel function of EC as a major SL source of circulatory SL species, which determines the SL content of organs (the lung and the liver) and the response of liver hepatocytes to chemical injury.

The SPT enzyme complex catalyzes the first committed step in the de novo SL biosynthetic pathway. This step is highly regulated by ORMDL proteins and other poorly understood mechanisms (*Cantalupo et al., 2015*; *Linn et al., 2001*; *Siow et al., 2015*). Since the vascular EC line blood vessels in all organ systems, we developed and analyzed a mouse genetic model in which S*ptlc1*, encoding the non-redundant subunit of SPT complex (SPTLC1) is disrupted in a tissue-specific manner using tamoxifen-induced gene deletion. This allowed us to precisely disrupt de novo SL synthesis in EC in a temporally manner. Our results revealed that lack of SPTLC1 protein in EC results in delayed vascular development. This phenotype was caused by reduced EC proliferation and tip cell formation in early angiogenic phases. In contrast, organotypic specialization of the developing vasculature was unaffected. This phenotype some similarities with the loss of VEGFR2 in the endothelium (*Zarkada et al., 2015*). In addition, in the OIR model, where pathological neovascularization is driven primarily by VEGF, inactivation of SPT activity in EC resulted in larger vaso-obliterated zones and reduced vascular tuft formation, again suggesting impaired VEGF signaling. Furthermore, intraocular administration of VEGF-A revealed that *Sptlc1* KO EC show attenuated response. Our in vitro studies in HUVEC demonstrated that SPT activity is necessary to maintain efficient VEGF signaling and its downstream substrate activation via lipid rafts. Supplementation of SL such as dhSph, Sph, and Cer is sufficient to rescue the inhibitory effects caused by Myriocin pretreatment, implicating active metabolic turnover of lipid raft constituents such as GM1. Our conclusion agrees with recent publications which described that the EC deletion of *Sptlc2* gene impaired VEGFR2-Akt signaling in vitro in adult arterial vessels

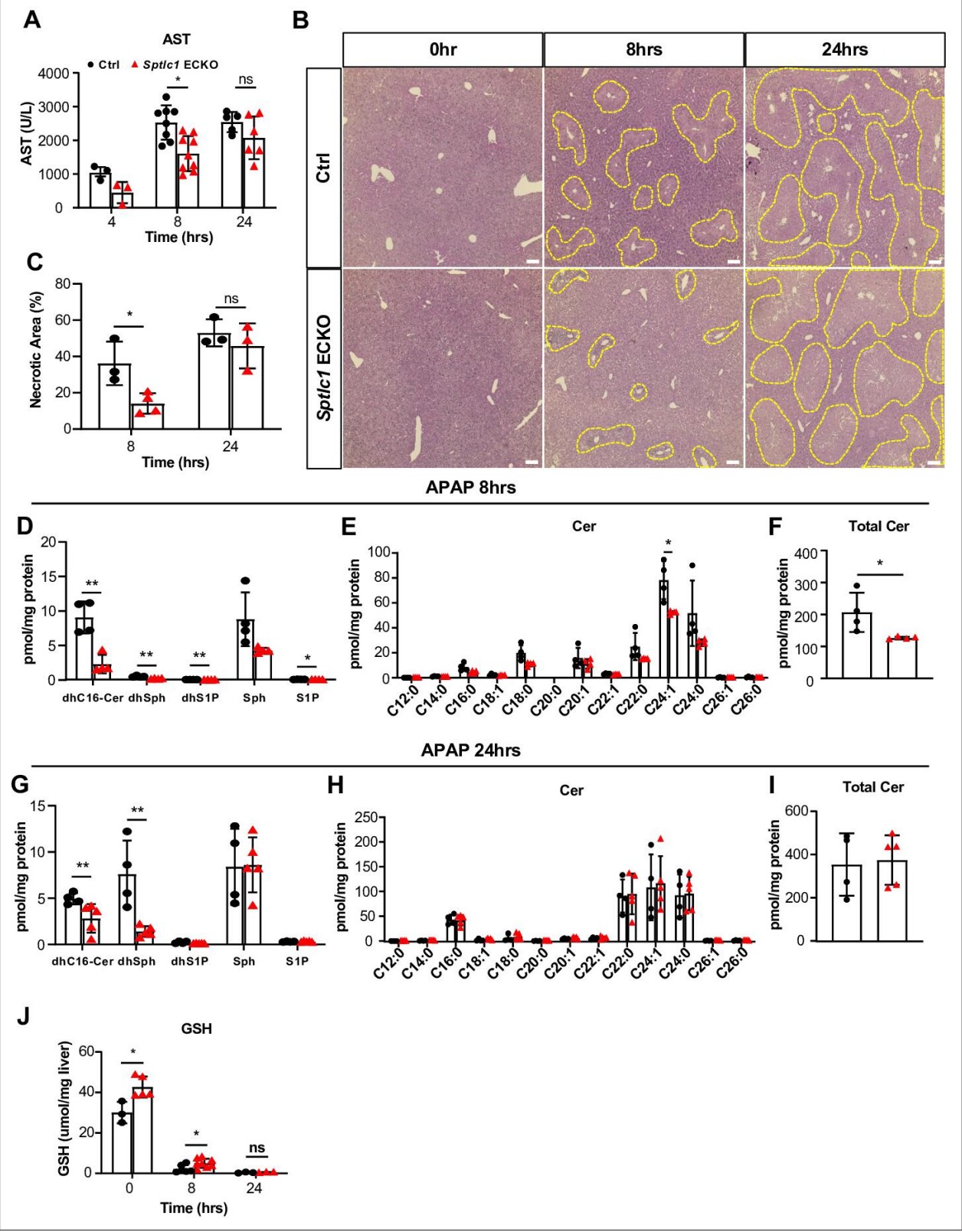

**Figure 9.** *Sptlc1* ECKO mice are protected against APAP-induced hepatoxicity. AST levels were measured from plasma samples of Ctrl and *Sptlc1* ECKO mice after 4, 8, and 24 hr of APAP injection (n=3–8 for Ctrl in each time point, n=3–9 for *Sptlc1* ECKO in each time point). (**B**) Liver sections from Ctrl and *Sptlc1* ECKO mice after 0, 8, and 24 hr of APAP administration were stained with hematoxylin and eosin. Representative images were shown and dotted lined areas indicate necrotic region. The percentage of necrotic area in total imaging area was quantified in (**C**) (n=3–5 for Ctrl in each time point, n=3–5

*Figure 9 continued on next page*

*Figure 9 continued*

for *Sptlc1* ECKO in each time point. each dot represents average of five images from one mouse). SL content in Ctrl and *Sptlc1* ECKO liver after 8 hr of APAP injection was measured by LC-MS/MS, including dhC16-Cer, dhSph, dhS1P, Sph and S1P (**E**), Cer with different fatty acyl chain lengths (**F**), total Cer content (**G**) (n=4 for Ctrl, n=4 for *Sptlc1* ECKO). SL content in Ctrl and *Sptlc1* ECKO liver after 24 hr of APAP injection was measured by LC-MS/MS, including dhC16-Cer, dhSph, dhS1P, Sph, and S1P (**H**), Cer with different fatty acyl chain length (**I**), total Cer content (**J**) (n=4 for Ctrl, n=5 for *Sptlc1* ECKO). (**D**) GSH levels were measured from liver samples of Ctrl and *Sptlc1* ECKO mice after 0, 8, and 24 hr of APAP injection (n=3–6 for Ctrl in each time point, n=3–8 for *Sptlc1* ECKO in each time point). Values of lipidomic results can be accessed in *Figure 9—source data 1*. Data are expressed as mean ± SD. Statistical significance was determined by unpaired t test. *p<0.05. **p<0.01. Scale bar, 1 mm.

The online version of this article includes the following source data for figure 9:

**Source data 1.** Levels of dhC16-Cer, dhSph, dhS1P, Sph, S1P, Cer, and from liver lysates of Ctrl and *Sptlc1* ECKO mice treated with APAP for 8 and 24 hrs.

---

(*Cantalupo et al., 2020*) and that lipid rafts enhance VEGFR2 signal transduction during angiogenesis (*Zabroski and Nugent, 2021*).

In addition to VEGFR2, endothelial nitric oxide synthase (eNOS) resides in SL-rich membrane domains such as lipid rafts or caveolae, suggesting that SPTLC1 loss of function could lead to modulation of eNOS activity. However, eNOS activity in arterial endothelium was elevated in *Sptlc2* ECKO adult mice (*Cantalupo et al., 2020*). In our *Sptlc1* ECKO adult mice, eNOS expressions and activation in lung EC was not altered (*Figure 1C* and *Figure 1—figure supplement 4*). We postulate that the discrepancy could results from isoform-selective functions between SPTLC1 and SPTLC2. Alternatively, stage-specific function of the SPT enzyme (early postnatal deletion in our study vs. adult stage deletion) or tissue-specific roles (retina EC vs. aorta EC) could explain these divergent effects of SPTLC1 and SPTLC2 on eNOS.

The decreased vascular development phenotype observed in *Sptlc1* ECKO mice does not resemble the hypersprouting phenotype nor is there down-regulation of BRB specific genes in mice that lack S1P receptors in EC (*Jung et al., 2012*; *Yanagida et al., 2020*). Mice lacking both sphingosine kinases (Sphk1 and Sphk2) in the endothelium did not display alterations in retinal vascular development (*Figure 3—figure supplement 2A, B*), suggesting that EC-derived S1P is not involved in this process. Furthermore, 50% reduction of plasma S1P observed in *Apom* KO mice, similar to what we observed in the plasma of *Sptlc*1 ECKO mice, exhibit normal developmental angiogenesis in the retina (*Figure 3—figure supplement 2C, D*). These findings suggest that reduced plasma S1P in *Sptlc1* ECKO mice is not the cause of the vascular defects seen in *Sptlc1* ECKO mice. Hence, we reasoned that membrane functions of SL metabolites, rather than GPCR-dependent signaling functions of S1P are involved.

Extracellular SL carried by lipoproteins and albuminare taken up by cells and metabolized to generate sphingosine (*Nilsson and Duan, 2006*). SL can also be degraded extracellularly, followed by cellular uptake of sphingosine (*Kono et al., 2006*). However, it is not known if SL derived from the de novo pathway are transferred between various cells and tissues. A previous report documented that liver specific KO of *Sptlc2* reduces plasma SM and Cer, while Sph, S1P and dhS1P are unaffected (*Li et al., 2009*). Our findings describe a novel metabolic flux of SL metabolites from EC to the circulatory system and peripheral tissues Moreover, SL species were rapidly reduced in RBC and plasma upon *Sptlc1* deletion in EC. The reduction of S1P levels in lipoprotein-free fraction of plasma but not in the HDL fraction of *Sptlc1* ECKO mice suggest that EC-derived S1P in the plasma interacts first with albumin, followed by stable interaction with ApoM on HDL.

A pivotal question raised from our results is how various SL species produced in the EC are transferred into plasma and circulatory cells. It has been reported that EC contribute to plasma S1P levels via SPNS2, an S1P transporter (*Hisano et al., 2012*), which raises the possibility that reduced circulatory SL is due to alterations in EC-derived S1P. We believe that this is not the case because loss of *Spns2* led to only minor reduction of plasma S1P levels but not in whole blood (*Hisano et al., 2012*; *Mendoza et al., 2012*). In addition, RBC S1P transporter (encoded by *Mfsd2b* gene) knockout mice show S1P accumulation in RBC and reduction in plasma (*Vu et al., 2017*). Our results from *Sptlc1* ECKO mice shows clear differences from the studies which characterized EC and RBC S1P transporter knockout mice. Having ruled out EC secretion of S1P as a possible mechanism, we next considered direct release of dhSph and Cer from EC into circulation, since such SL metabolites were reduced in *Sptlc1* ECKO plasma and RBC. While the existence of transporters for dhSph and Cer species have not been described, we do not believe that sphingoid bases and Cer species are directly secreted

from EC into circulation. However, alternative ways of transfer may exist. One possibility is through direct contact of EC with lipoproteins and/or RBC. Given that acid sphingomyelinase and ceramidase activities have been reported in the plasma and RBC (*López et al., 2012*; *Xu et al., 2010*), complex SL containing both reduced and unreduced sphingoid bases on the outer leaflet of the EC plasma membrane could be metabolized into Cer and sphingoid bases. SL species such as Sph can be taken up and further metabolized by circulating hematopoietic cells including RBC (*Nguyen et al., 2021*). An alternative mechanism is that EC release of circulating extracellular vesicles (EV) or exosomes from plasma membranes into circulation. It has been reported that Cer, SM, and glycosphingolipids are enriched in EV (*Llorente et al., 2013*; *Verderio et al., 2018*; *Akawi et al., 2021*). It is thus possible that EV derived from EC can bind to RBC or lipoproteins and SL can be taken up and metabolized.

In addition, we observed reduced SL levels in non-EC and peripheral tissues from *Sptlc1* ECKO mice. Specific organs and tissues may have different routes for SL transfer based on the unique barrier properties of the organ-specific endothelium. In the liver, EC form fenestrated capillaries with inter-cellular gaps and discontinuous basement membrane. This allows plasma and macroparticles, to cross EC barrier efficiently and access hepatocytes (*Hennigs et al., 2021*). SL exchange could therefore occur in the liver due to the porous nature of the liver sinusoidal endothelium. We postulate that the SL metabolic flux from EC to the plasma followed by hepatocyte uptake could explain the reduced liver SL content observed in *Sptlc1* ECKO mice. In the case of lung, pulmonary EC are non-fenestrated with tight junctions in between and serve as a semipermeable barrier separating circulation from lung interstitial spaces. To transport materials through lung capillaries, transcytosis acts as the primary route (*Jones and Minshall, 2020*). It is known to be mediated by caveolae, a critical vesicular structure, which are plasma membrane invaginations enriched in SL and sterols (*Harder and Simons, 1997*). We speculate that the SL flux in the lung could be mediated by this process which is affected by loss of EC *Sptlc1* gene. This mechanism could account for the reduced lung SL species in *Sptlc1* ECKO mice. In contrast, caveolae-mediated transcytosis is minimal in the CNS EC which effectively compartmentalizes metabolism of lipids between the circulation and the CNS parenchyma. Our results show that retina SL species were unaffected by lack of de novo SL biosynthesis in EC. Thus, the strict barrier function of the CNS EC ensures compartmentalization of SL metabolism between the neural retina and circulatory/vascular elements.

Finally, we show that *Sptlc*1 ECKO liver is protected against acetaminophen-induced hepatotoxicity. Increased GSH levels in *Sptlc1* ECKO liver suggests that glutathione synthesis, a primary detoxification agent of APAP, may be involved in mediating the protective phenotype. It remains to be determined whether hepatic SL levels regulate GSH levels directly or indirectly. We speculate that lack of SPT activity in EC may allow the unused serine to be converted to GSH since serine is a source of glycine and cysteine used to synthesize glutathione (*Zhou et al., 2017*). Nevertheless, our results show that SL provided by EC is needed for APAP-induced hepatotoxicity. Recently, Cer species increases has been reported to be involved in drug induced liver injury models in mice (*Li et al., 2020*). Our liver lipidomic results further suggest a delayed accumulation of Cer in *Sptlc1* ECKO mice upon APAP treatment (*Figure 8E and H*), which correlates with its protective phenotype at early time points. It is worth noting that administration of Fumonisin B1 (FB1), a Cer synthase inhibitor, does not alleviate APAP-induced liver damage (*Park et al., 2013*). FB1 pretreatment reduces Cer content in the liver while accumulating dhSph, which is in contrast of what we observed in *Sptlc1* ECKO liver. Future experiments with myriocin, a direct inhibitor of SPT complex, would be warranted to determine the potential role of dhSph during APAP-induced liver injury.

In conclusion, our studies suggest that de novo SL synthesis pathway in EC is needed for normal and pathological angiogenesis. This is partially due to impaired VEGF responsiveness via modulation of lipid raft formation (*Figure 10A*). Secondly, we also reveal a novel function of the endothelium as a source of SL metabolites in circulation, lung and liver but not the retina (*Figure 10B*). This EC-derived SL flux mechanism may be impaired in EC dysfunction, a common mechanism in many cardiovascular and cerebrovascular diseases.

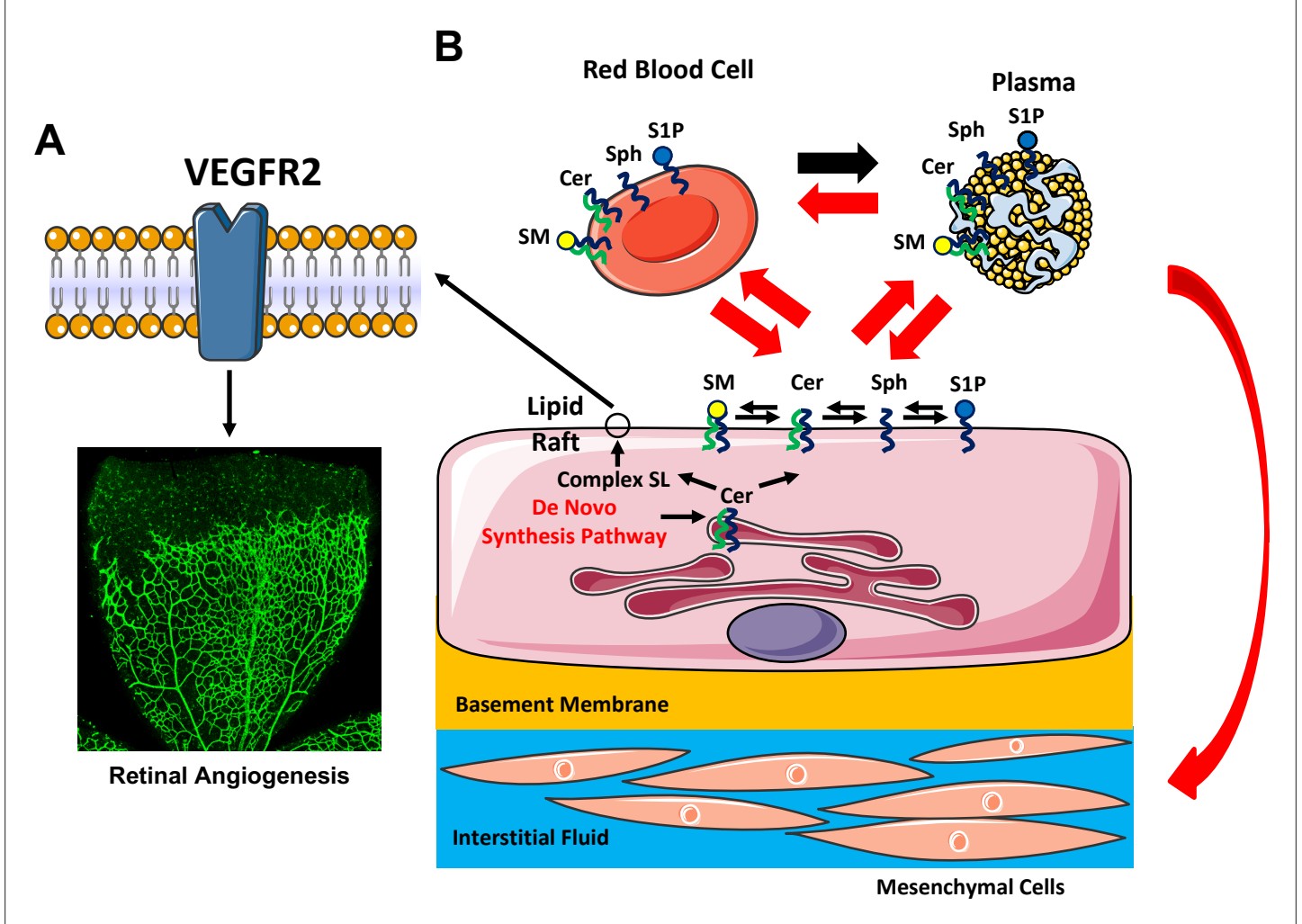

**Figure 10.** Graphic abstract summarizing major findings of the study. (**A**) Abrogation of SPT activity in EC leads to reduction of SL content, causing EC-intrinsic effects including disruption of lipid rafts. This alteration impairs VEGF-responsiveness, resulting in delayed developmental and pathological angiogenesis. (**B**) Our studies also reveal that loss of SPT activity in EC exhibit EC-extrinsic effects, where rapid reduction of several SL metabolites in plasma, red blood cells and peripheral organs (lung and liver) but not in the retina, part of the CNS.

## Materials and methods
### Mouse strains

*Sptlc1*fl/fl mice were generated as previously described (*Alexaki et al., 2017*). Endothelial specific deletion of *Sptlc1*(*Sptlc1* ECKO) were generated by intercrossing *Sptlc1*fl/fl mice with *Cdh5*-Cre-ERT2 strain (*Sörensen et al., 2009*). *Sptlc1* ECKO mice used in the experiments were *Sptlc1*fl/fl carrying one *Cdh5*-ERT2-Cre allele. Controls were littermate *Sptlc1*fl/fl mice. To identify *Sptlc1*fl/fl allele and WT, the following primers were used: 5'-GGG TTC TAT GGC ACA TTT GGT AAG-3' (forward primer), and 5'-CTG TTA CTT CTT GCC AGT GGA C-3' (reverse primer), generating products of 350 bp for WT and 425 bp for *Sptlc1*fl/fl allele (*Alexaki et al., 2017*). The Cdh5-Cre-ERT2 allele was detected by PCR with the following primers: 5'-TCC TGA TGG TGC CTA TCC TC-3' (forward primer), and 5'-CCT GTT TTG CAC GTT CAC CG-3' (reverse primer), generating a product of 550 bp. The recombined *Sptlc1* knock-out allele was identified by PCR using 5'-CAG AGC TAA TGG AAA GGT GTC-3' (forward primer) and 5'-CTG TTA CTT CTT GCC AGT GGA C-3' (reverse primer), generating a product of 315 bp. To induce the deletion of *Sptlc1* in *Figures 1, 3–5*, *Figure 6A-C*, *Figure 7*, *Figure 8*, 6- to 12-week-old mice were given 2 mg tamoxifen (Sigma) dissolved in corn oil (Sigma) via intraperitoneal injection for 3 consecutive days. Experiments in the study were performed on animals at least 4 weeks post tamoxifen treatment other than animals used in *Figure 6A and B*. For *Figure 2* to *Figure 4B-G*

and *Figure 6D-J*, neonates from *Sptlc1*ECKO strain were given 50 µg tamoxifen via intraperitoneal injection for 3 consecutive days from P1 to P3. Both male and female animals were used for the experiments. All animal experiments were approved by the Boston Children's Hospital Institutional Animal Care and Use Committee (IACUC).

## Cell line

Human umbilical vein endothelial cells (HUVEC) were purchased from Lonza and cultured in M199 medium (Corning) supplemented with 10% FBS, penicillin–streptomycin, endothelial cell growth factor from sheep brain extract, and 5 U/ml heparin on gelatin–coated dishes in a 37 °C incubator with 5% CO2. Passage 4 – passage 7 cells were used for experiments. Endothelial markers including VE-Cad, CD31, and ERG were used for cell characterization by immunofluorescence.

Detailed cell authentication information of HUVEC can be acquired from Lonza. In brief, routine characterization of HUVEC includes morphological observation throughout serial passages and all lots produced after 01 May 2010 stain at least 90% double-positive for endothelial cell markers CD31/ CD105 and negative for alpha smooth muscle actin marker at passage 4. Pre-screened HUVEC are guaranteed to test positive for angiogenesis and vascular cell health markers, eNOS, Tie-2, VEGFr2, and Axl in the first passage out of cryopreservation (From Lonza).

Upon purchase, all cells are performance assayed and test negative for HIV-1, mycoplasma, hepatitis-B, hepatitis-C, bacteria, yeast and fungi. Cell performance and characterization are measured after recovery from cryopreservation.

## Mouse EC isolation

Mouse lung EC were isolated from three to four pairs of lungs dissected from control or *Sptlc1* ECKO adult mice 4 weeks post tamoxifen treatment. Freshly isolated lung tissues were minced with scissors and allowed to digest at 37 °C in Liberase (0.6 U/mL, Sigma) in HBSS for 45 min. Samples were further subjected to mechanical disruption by gentleMACS Octo Dissociator (Miltenyi Biotec) and filtration through fine steel mesh (40 µm, Falcon). Cells were washed twice with HBSS containing 0.5% Fatty Acid Free BSA. RBC were removed by ACK lysis buffer (Thermo Fisher). Cells were incubated with CD45 micro beads (Miltenyi Biotec) at 4 °C for 15 min to deplete CD45$^+$ cells. Remaining cells were incubated with CD31 micro beads (Miltenyi Biotec) at 4 °C for 15 min to enrich EC. The purity of each population was determined by flow cytometry.

Cells were stained with stained with phycoerythrin (PE)-conjugated anti-mouse CD31 (BioLegend, 1:200), allophycocyanin (APC)-conjugated anti-mouse CD45 (BioLegend, 1:200) in blocking solution (0.25% FAF-BSA in HBSS) with anti-CD16/32 (BioLegend, 1:200) for 15 min on ice. Stained cells were washed with HBSS containing 10% serum twice and analyzed by BD FACSCalibur Flow Cytometer (BD Biosciences). CD31 and CD45 were gated as shown in *Figure 1B*.

## Western blotting

Isolated cells from EC isolation procedure and HUVEC were extracted with lysis buffer (50 mM, Tris·HCl (pH 7.4), 0.1 mM EDTA, 0.1 mM EGTA, 1% Nonidet P-40, 0.1% sodium deoxycholate, 0.1% SDS, 100 mM NaCl, 10 mM NaF, 1 mM sodium pyrophosphate, 1 mM sodium orthovanadate, 25 mM sodium β-glycerophosphate, 1 mM Pefabloc SC, and 2 mg/mL protease inhibitor mixture (Roche Diagnostics)). Protein concentration was determined by Dc Protein Assay kit (Bio-Rad). 20 to 50 µg of total proteins were analyzed by SDS/PAGE and transferred onto a nitrocellulose membrane. Blots were blocked with 5% non-fat milk in TBS. Blocked blots were incubated with primary antibodies including SPTLC1 (Santa Cruz, 1:1000), eNOS (BD Biosystem, 1:1000), β-actin (Sigma, 1:5000), phospho-eNOS (Cell Signaling Technology, 1:1000), phospho-ERK (Cell Signaling Technology, 1:2000), ERK (Cell Signaling Technology, 1:2000) at 4 °C overnight. Blots were washed three times with TBSt for 5 min before adding the HRP-conjugated secondary antibody (Millipore, 1:5000). After another three times wash with TBSt, blots were developed with ECL western blotting substrate (Thermo Fisher) and imaged by Azure 600 imaging system (Azure Biosystems).

## SL measurement of isolated cells

Isolated cells from EC isolation procedure were counted and 2–5x10$^6$ cells were subjected to HPLC-mass spectrometry (MS/MS) by the Lipidomics Core at the Stony Brook University Lipidomic Core

(Stony Brook, NY) using TSQ 7000 triple quadrupole mass spectrometer, operating in a multiple reaction monitoring-positive ionization mode as described (*Bielawski et al., 2010*).

## Retina immunohistochemistry in flat mounts

Eyeballs from Ctrl and *Sptlc1* ECKO mice were enucleated and fixed in 4% Paraformaldehyde (PFA) in PBS at room temperature for 20 min. Retinas were isolated under dissection microscopy, permeabilized in 0.5% Triton X-100 in PBS (PBSt) at room temperature for 30 min and blocked with 1% Bovine Serum Albumin (BSA) in PBSt at room temperature for 30 min. Primary antibodies were added to the sections with indicated dilution in PBSt (NG2 (Sigma, 1:200), Phsopho-histone H3 (BioLegend, 1:200), ESM1 (R and D system, 1:200), LEF1 (Cell Signaling Technology, 1:200), TFRC (Novus Biologicals, 1:200), CLDN5 (Invitrogen, 1:200), MFSD2A (A kind gift from David Silver, 1:200), Alexa Fluor 647 isolectin GS-IB4 conjugate (Invitrogen, 1:500), Alexa Fluor 488-conjugated rabbit monoclonal anti- ERG (Abcam, 1:200), Alexa Fluor 647-conjugated rabbit monoclonal anti- ERG (Abcam, 1:200), Cy3-conjugated mouse monoclonal anti-α-smooth muscle actin (Sigma, 1:500)) at 4 °C overnight. Retinas were further washed three times with PBS for 10 min at room temperature and incubated with fluorescent conjugated secondary antibodies with 1:500 dilution in PBSt for 2 hr at room temperature. Stained retinas were washed three times with PBS for 10 min at room temperature and mounted using Fluoromount-G slide mounting medium (Southern Biotech). Flat Mount Retina Slides were proceeded to image with confocal microscopy (Zeiss).

## Oxygen induced retinopathy

Retinal neovascularization was observed using the OIR model as published (*Smith et al., 1994*). In brief, neonates together with their mother were exposed to 75% oxygen from P7 to P12. Upon returning to room air (20% $O_2$) at P12 the relative hypoxia induced neovascularization over the period P12 through P17 in these mice, with P17 as the maximum of neovascularization. For experiments in *Figure 4H–K*, 100 μg tamoxifen was administrated via intraperitoneal injection at P12. Eyeballs were collected at P10, P12 or P17 for immunofluorescence (See section above). Retinal neovascularization and vaso-obliteration were quantified using Image J as previously reported (*Stahl et al., 2009*).

## Intravitreal injection of VEGF-A

Intravitreal injection of mouse recombinant VEGF-A165 (BioLegend) was performed in P6 pups. Pups were first anesthetized by using crushed ice. Eye lids were carefully separated by 30 G needle and eyeballs were lifted for injection.1 μL of VEGF-A165 at a concentration of 50 ng/μl or 1 μL of saline were injected into the vitreous humor using a Nanofil syringe (World Precision Instrument). Pups were placed onto a heating pad and returned to mom's care. After 4 hr, eyeballs were collected and processed for retina immunohistochemistry.

## SL supplementation to HUVEC

A total of 100 nM Myriocin (Cayman) were treated to HUVEC for 72 hr in complete M199 medium. They were starved for 24 hr in M199 medium supplemented with 10% charcoal stripped FBS and 4 mg/mL FAF-BSA. 6 hr prior to VEGF treatment, dhSph, Sph and C16:0 Cer (300 nM) were added to the medium. VEGF (50 ng/mL) were then treated for 5 min and protein extracts were harvested in lysis buffer.

## Lipid raft staining by Cholera Toxin B

Lipid rafts were stained by Vybrant Lipid Raft labeling kit (Invitrogen) followed by manufacturer's instruction.

## SL measurement of liver and lung

Tissues were homogenized in 20 mM Tris-HCl pH 7.5 with tissue homogenizer (Fisher Scientific). Protein concentration was determined by Dc Protein Assay kit (Bio-Rad). Two to 6 mg of total protein were used for HPLC-MS/MS by the Lipidomics Core at the Stony Brook University Lipidomic Core (https://osa.stonybrookmedicine.edu/research-core-facilities/bms Stony Brook, NY) using TSQ 7000 triple quadrupole mass spectrometer, operating in a multiple reaction monitoring-positive ionization mode as described (*Bielawski et al., 2010*).

## S1P measurement in plasma and RBC

Plasma S1P and RBC S1P in *Figure 6A–C* was extracted as previously described (*Frej et al., 2015*) with minor modification. Ten μL of plasma or $5 \times 10^7$ of RBC were first diluted to 100 μL with TBS Buffer (50 mM Tris-HCl pH 7.5, 0.15 M NaCl). S1P was extracted by adding 100 μL precipitation solution (20 nM D7-S1P in methanol) followed by 30 s of vortexing. Precipitated samples were centrifuged at 18,000 rpm for 5 min and supernatant were transferred to vials for UHPLC-MS/MS analysis (see below). C18-S1P (Avanti Lipids) was dissolved in methanol to obtain a 1 mM stock solution. Standard samples were prepared by diluting the stock in 4% fatty acid free BSA (Sigma) in TBS to obtain 1 mM and stored at –80 °C. Before analysis, the 1 mM S1P solution was diluted with 4% BSA in TBS to obtain the following concentrations: 0.5 mM, 0.25 mM, 0.125 mM, 0.0625 mM, 0.03125 mM, 0.0156 mM, and 0.0078 mM. S1P in diluted samples (100 μL) were extracted with 100 μL of methanol containing 20 nM of D7-S1P followed by 30 s of vortexing. Precipitated samples were centrifuged at 18,000 rpm for 5 min and the supernatants were transferred to vials for mass spectrometric analysis. The internal deuterium-labeled standard (D7-S1P, Avanti Lipids) was dissolved in methanol to obtain a 200 nM stock solution and stored at –20 °C. Before analysis, the stock solution was diluted to 20 nM for sample precipitation. The samples were analyzed with Q Exactive mass spectrometer coupled to a Vanquish UHPLC System (Thermo Fisher Scientific). Analytes were separated using a reverse phase column maintained at 60 °C (XSelect CSH C18 XP column 2.5 mm, 2.1 mm X 50 mm, Waters). The gradient solvents were as follows: Solvent A (water/methanol/formic acid 97/2/1 (v/v/v)) and Solvent B (methanol/acetone/water/formic acid 68/29/2/1 (v/v/v/v)). The analytical gradient was run at 0.4 mL/min from 50–100% Solvent B for 5.4 min, 100% for 5.5 min, followed by one minute of 50% Solvent B. A targeted MS2 strategy (also known as parallel reaction monitoring, PRM) was performed to isolate S1P (380.26 m/z) and D7-S1P (387.30 m/z) using a 1.6 m/z window, and the HCD-activated (stepped CE 25, 30,50%) MS2 ions were scanned in the Orbitrap at 70 K. The area under the curve (AUC) of MS2 ions (S1P, 264.2686 m/z; D7-S1P, 271.3125 m/z) was calculated using Skyline (*MacLean et al., 2010*). Quantitative linearity was determined by plotting the AUC of the standard samples (C18-S1P) normalized by the AUC of internal standard (D7-S1P); (y) versus the spiked concentration of S1P (x). Correlation coefficient ($R^2$) was calculated as the value of the joint variation between x and y. Linear regression equation was used to determined analyte concentrations.

## Plasma and RBC collection

For non-terminal blood collection, blood was collected from submandibular vein via cheek punch. For terminal blood collection, mice were euthanized with $CO_2$. Blood was recovered via vina cava. Blood samples were collected in tubes containing 1–5 μl 0.5 M EDTA depending on sample volume. Samples were centrifuged at 2000 g for 10 min. Plasma samples were collected and stored at –80 °C. RBC were washed twice with HBSS and cell number was counted by hemocytometer. $10^9$ cells were pelleted by centrifugation at 2000 g for 10 min and store at –80 °C for SL measurement.

## SL measurement in plasma and retina

Retinas were placed on dry ice for their extraction. Afterwards, 600 μl of ice cold (Freezer for 1 hr, kept on ice while processing outside) LC-MS methanol was added to each sample, followed by 10 μl of the deuterated sphingolipid labeled internal standards. Samples were then vortexed and allowed to equilibrate on ice for 15 min. This was followed by two 15 min sonication steps in an ice bath. Next, samples were centrifuged at 12,000 *g* for 15 min at 6°C. Finally, 120 μl of the supernatant were transferred to an LC-MS amber vial equipped with a 150 μl insert for LC-MS injection. A pool of 25 μL of all samples was used as QC of the injection. Samples were injected on an Acquity UPLC system coupled to a Xevo TQ-S mass spectrometer (Waters, Milford, MA) as previously described (*Akawi et al., 2021*). For plasma analyses, samples were thawed at 4°C in the refrigerator. Then, samples were extracted as previously described (*Akawi et al., 2021*). Briefly, a volume of 10 μl of the sphingolipid internal standard mix containing labeled internal standards. Samples were then vortexed for 10 s and equilibrated with the internal standard at room temperature. Then, a volume of 250 μl LC-MS methanol was added to each sample, followed by a 10-s vortex period and sonication for 15 min in an ultrasound bath with ice. Next, samples were centrifuged at 12,000 *g* for 15 min at 6°C. Finally, 120 μl of the supernatant were transferred to an LC-MS amber vial equipped with a 150 μl glass insert. A pool of

25 µL of all samples was used as QC of injection. Samples were then analyzed using the same method as described for the retinas.

## APAP-induced liver injury

Mice were fasted for 16 hr. APAP (Sigma) was dissolved in PBS and administered to the mice via intra-peritoneal injection (300 mg/kg). Food and water were given ad lib after treatment. Plasma and liver samples were collected at indicated time for further analysis. Liver function was measured using AST activity assay kits (Sigma) according to the manufacturer's instructions. Liver GSH levels were measured using glutathione assay kits (Cayman Chemical) according to the manufacturer's instructions. A lobe of liver was fixed in 4% PFA and processed for paraffin embedding. Sections were stained with hematoxylin and eosin (H & E). Images were captured by Axioskop 2 mot Plus microscope (5 x/0.12).

## Confocal microscopy and image processing

Images were acquired using a LSM810 confocal microscope (Zeiss) equipped with an EC Plan-Neofluar 10 x/0.3, a Plan-Apochromat 20 x/0.8 or a Plan-Apochromat 40 x/1.4 Oil DIC objective. Images were taken using Zen2.1 software (Zeiss) and processed and quantified with Fiji (NIH). Figures were assembled using Affinity Photo and Windows Office Powerpoint.

## Quantification of images

Vascular density quantification (*Figure 2B, J and K*, *Figure 3—figure supplement Figure 3—figure supplements 1 and 2*, *Figure 4C and F*).

Tile-scanned images of flat mounted retinas stained with IB4 were used. IB4$^+$ area was quantified as a percent of area of region of interest via Vessel Analysis plugin on Fiji (NIH). For 'total vascular density', z-stack tile-scanned images containing all three plexuses were acquired and used for quantification. For inner plexus density and superficial plexus density, z-stack tile-scanned images containing specified plexus were acquired and used for quantification. IB4$^+$ area was quantified as a percent of total retina area as the vasculature has developed at P10 and P15. Value from a single retina image is shown as a dot.

Vessel outgrowth quantification (*Figure 2C*).

Tile-scanned images of flat mounted retinas stained with IB4 were used. The distance between the vascular front and the central optic nerve was quantified as a percent of the retinal radius. The mean of values from 4 different leaves of retina is shown as a dot.

Pericyte coverage quantification (*Figure 2E*).

Tile-scanned images of flat mounted retinas stained with IB4/NG2 were used. NG2$^+$ fraction signal is divided by vascularized area (IB4$^+$). Value from a single retina image is shown as a dot.

Proliferating cell quantification (*Figure 2G*).

Tile-scanned images of flat mounted retinas stained with IB4/pHH3 and ERG were used. ERG$^+$; pHH3$^+$ cells within the vasculature (IB4$^+$ area) were manually counted. Value from a single retina image is shown as a dot.

Tip cell quantification (*Figure 2H*).

Tile-scanned images of flat mounted retinas stained with IB4/pHH3 and ERG were used. ERG$^+$; ESM1$^+$ cells were manually counted. Value from a single retina image is shown as a dot.

Positive area of BRB associated gene quantification (*Figure 3B, C, E and F*).

Tile-scanned images of flat mounted retinas stained with IB4 and either LEF-1, TFRC, MFSD2A and CLDN5 were acquired. The area double-positive for either MFSD2A, TFRC or LEF-1 and IB4 was quantified as a percent of IB4$^+$ area. Value from one representative retinal leave is plotted as a dot.

Vaso-obliteration (VO) area quantification (*Figure 4D, G and K*).

Tile-scanned images of flat mounted retinas stained with IB4/SMA were used. VO area was determined by avascularized area divided by total retinal area using Affinity Photo software.

Neovascularization (NV) area quantification (*Figure 4J*).

Tile-scanned images of flat mounted retinas stained with IB4/SMA were used. NV area was determined by manually circling neovascularized area (IB4$^+$/SMA$^+$ areas with abnormal shape than regular vasculature) and divided by total retinal area using Affinity Photo software.

Vascular front EC number quantification (*Figure 5B*).

Tile-scanned images of flat mounted retinas stained with IB4 and ERG were used. Vascular front areas were manually lassoed based on IB4 signal. Numbers of ERG+ cells were manually counted. Values from a single retina image is shown as a dot.

ESM expressions in capillary plexus quantification (*Figure 5C*).

Tile-scanned images of flat mounted retinas stained with IB4, ERG and ESM1. Vascular front areas were manually lassoed. Capillary plexus regions were defined as areas outside of vascular front. Numbers of ESM1+/ERG+ cells in both regions were manually counted. Value from a single retina image is shown as a dot.

Necrotic area quantification (*Figure 9C*).

Necrotic areas were determined manually based on H and E staining and hepatocyte nuclear morphology and further divided by total area of the image. Mean value from 5 representative images is shown as a dot.

## Statistics

For datasets containing exactly two groups, an unpaired two-tailed Student's t test was used to determine significant differences. In *Figure 1—figure supplement 1B*, *Figure 7A and B*, one-way ANOVA followed by Bonferroni's post hoc test was used to determine significant differences at all time points. p Value less than 0.05 was statistically significant.

## Acknowledgements

We thank Lipidomics Core Facility of Stony Brook University Medical Center for the analytical work in MS. This work was supported in part by the NIH grants (R35-HL135821 and R01EY031715 to TH), Intramural Research Programs of the National Institutes of Health, National Institute of Diabetes and Digestive and Kidney Disease to RLP, National Institutes of Health, and American Heart Association Postdoctoral Fellowship (18POST33990339) to AK.

## Additional information

### Competing interests

Lois E Smith: Reviewing editor, eLife. The other authors declare that no competing interests exist.

### Funding

| Funder | Grant reference number | Author |
|---|---|---|
| American Heart Association | Postdoctoral Fellowship | Andrew Kuo |
| National Heart, Lung, and Blood Institute | R35 | Timothy Hla |
| National Eye Institute | R01 | Timothy Hla |
| American Heart Association | 18POST33990339 | Andrew Kuo |
| National Heart, Lung, and Blood Institute | HL135821 | Timothy Hla |
| National Eye Institute | EY031715 | Timothy Hla |

The funders had no role in study design, data collection and interpretation, or the decision to submit the work for publication.

### Author contributions

Andrew Kuo, Conceptualization, Data curation, Formal analysis, Investigation, Methodology, Writing - original draft, Writing – review and editing; Antonio Checa, Data curation, Formal analysis, Validation, Investigation, Methodology, Writing – review and editing; Colin Niaudet, Formal analysis, Investigation, Methodology, Writing – review and editing; Bongnam Jung, Formal analysis, Investigation,

Writing – review and editing; Zhongjie Fu, Resources, Software, Methodology, Writing – review and editing; Craig E Wheelock, Resources, Data curation, Formal analysis, Methodology, Writing – review and editing; Sasha A Singh, Resources, Data curation, Methodology, Project administration, Writing – review and editing; Masanori Aikawa, Resources, Project administration, Writing – review and editing; Lois E Smith, Resources, Supervision, Methodology, Writing – review and editing; Richard L Proia, Conceptualization, Resources, Supervision, Writing – review and editing; Timothy Hla, Conceptualization, Data curation, Supervision, Funding acquisition, Writing - original draft, Project administration, Writing – review and editing

### Author ORCIDs

Andrew Kuo http://orcid.org/0000-0002-7263-8658
Zhongjie Fu http://orcid.org/0000-0002-8182-2983
Craig E Wheelock http://orcid.org/0000-0002-8113-0653
Masanori Aikawa http://orcid.org/0000-0002-9275-2079
Lois E Smith http://orcid.org/0000-0001-7644-6410
Richard L Proia http://orcid.org/0000-0003-0456-1270
Timothy Hla http://orcid.org/0000-0001-8355-4065

### Ethics

This study was performed in strict accordance with the recommendations in the Guide for the Care and Use of Laboratory Animals of the National Institutes of Health. All of the animals were handled according to approved institutional animal care and use committee (IACUC) protocols (#19-10-4031R) of the Boston Children's Hospital. Every effort was made to minimize suffering.

### Decision letter and Author response

Decision letter https://doi.org/10.7554/eLife.78861.sa1
Author response https://doi.org/10.7554/eLife.78861.sa2

---

## Additional files

### Supplementary files

• MDAR checklist

### Data availability

All data generated or analyzed during this study are included in the manuscript and supporting files. Source data files have been provided for each figure.

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

# Appendix 1

### Appendix 1—key resources table

| Reagent type (species) or resource | Designation | Source or reference | Identifiers | Additional information |
|---|---|---|---|---|
| Genetic reagent (*M. musculus*) | Sptlc1fl/fl mice | *Alexaki et al., 2017* (PMID:28100772) | RRID:MMRRC_042294-UCD | |
| Genetic reagent (*M. musculus*) | Cdh5-Cre-ERT2 mice | *Sörensen et al., 2009* (PMID:19144989) | RRID:MGI:3848984 | |
| Genetic reagent (*M. musculus*) | Sphk1fl/fl mice | *Pappu et al., 2007* (PMID:17363629) | RRID:MMRRC_030038-UCD | |
| Genetic reagent (*M. musculus*) | Sphk2-/- mice | *Mizugishi et al., 2005* (PMID:16314531) | RRID:IMSR_JAX:019140 | |
| Genetic reagent (*M. musculus*) | Apom-/- mice | *Christoffersen et al., 2008* (PMID:18006500) | RRID:MGI:3772574 | |
| Cell line (Human) | Human eubmilical vein endothelial cells | Lonza | C2519A | |
| Antibody | Anti-CD31 MicroBeads (rat monoclonal) | Miltenyi Biotec | 130-097-418; RRID:AB_2814657 | |
| Antibody | Anti-CD45 MicroBeads (rat monoclonal) | Miltenyi Biotec | 130-052-301; RRID:AB_2877061 | |
| Antibody | (PE)-conjugated Anti-CD31 (rat monoclonal) | Biolegend | 102508; RRID:AB_312915 | For EC Isolation (1:250) |
| Antibody | (APC)-conjugated Anti-mouse CD45 (rat monoclonal) | Biolegend | 103112; RRID:AB_312977 | For EC Isolation (1:250) |
| Antibody | Anti-SPTLC1 (mouse monoclonal) | Santa Cruz Biotechnology | sc-374143; RRID:AB_10917035 | For Western Biotting (1:1000) |
| Antibody | Anti-phospho eNOS (rabbit monoclonal) | Cell Signaling Techonology | 9571; RRID:AB_329837 | For Western Biotting (1:1000) |
| Antibody | Anti-eNOS (rabbit polyclonal) | BD Biosciences | Cat #:610296; RRID:AB_397691 | For Western Biotting (1:1000) |
| Antibody | Anti-β-Actin (mouse monoclonal) | Sigma-Aldrich | Cat# A5316; RRID:AB_476743 | For Western Biotting (1:5000) |
| Antibody | Anti-rabbit IgG-HRP conjugated (goat polyclonal) | Abcam | Cat# ab205718; RRID:AB_2819160 | For Western Biotting (1:5000) |
| Antibody | Anti-mouse IgG-HRP conjugated | Millipore | AP-124P; RRID:AB_9045 | For Western Biotting (1:5000) |
| Antibody | Anti-ERK (rabbit polyclonal) | Cell Signaling Techonology | Cat# 9102; RRID:AB_330744 | For Western Blotting (1:2000) |
| Antibody | Anti-phospho ERK (rabbit monoclonal) | Cell Signaling Techonology | Cat# 9106; RRID:AB_331768 | For Western Blotting (1:2000) |
| Antibody | Anti-ApoM (rabbit monoclonal) | Abcam | ab91656; RRID:AB_204916 | For Western Blotting (1:2000) |
| Antibody | Anti-NG2 (rabbit polyclonal) | Sigma-Aldrich | AB5320; RRID: AB_91789 | For Immunofluorescence (1:200) |
| Antibody | Anti Phospho-histone H3 (Ser28, HTA28) (rat monoclonal) | Biolegend | 641001; RRID: AB_1227660 | For Immunofluorescence (1:200) |
| Antibody | Anti-ESM1 (goat polyclonal) | R&D Systems | AF1999; RRID: AB_2101810 | For Immunofluorescence (1:200) |

*Appendix 1 Continued on next page*

*Appendix 1 Continued*

| Reagent type (species) or resource | Designation | Source or reference | Identifiers | Additional information |
|---|---|---|---|---|
| Antibody | Anti-ERG (rabbit monoclonal) | Abcam | Cat# ab92513; RRID:AB_2630401 | For Immunofluorescence (1:500) |
| Antibody | Alexa Fluor 488-conjugated Anti- ERG (rabbit monoclonal) | Abcam | ab196374; RRID:AB_2889273 | For Immunofluorescence (1:200) |
| Antibody | Alexa Fluor 647-conjugated Anti- ERG (rabbit monoclonal) | Abcam | ab196149 | For Immunofluorescence (1:200) |
| Antibody | Anti-LEF1 (rabbit monoclonal) | Cell Signaling Techonology | 2230; RRID:AB_823558 | For Immunofluorescence (1:200) |
| Antibody | Anti-TFRC (rat monoclonal) | Novus Biologicals | NB100-64979; RRID: AB_962622 | For Immunofluorescence (1:200) |
| Antibody | Anti-CLDN5 (rabbit polyclonal) | Invitrogen | 34–1600; RRID: AB_2533157 | For Immunofluorescence (1:200) |
| Antibody | Anti-MFSD2A (rabbit polyclonal) | Gift from David Silver Lab | PMID: 23209793 | For Immunofluorescence (1:200) |
| Antibody | Anti-Actin, α-Smooth Muscle - Cy3 Antibody (mouse monoclonal) | Sigma-Aldrich | C6198; RRID:AB_476856 | For Immunofluorescence (1:200) |
| Antibody | Human VE-Cadherin Antibody (goat polyclonal) | R&D Systems | AF938; RRID:AB_355726 | For Immunofluorescence (1:200) |
| Chemical compound or drug | Tamoxifen | Sigma-Aldrich | T5648 | |
| Chemical compound or drug | Corn Oil | Sigma-Aldrich | C8267 | |
| Chemical compound or drug | Liberase TM Research Grade | Sigma-Aldrich | 5401127001 | |
| Chemical compound or drug | ACK lysis buffer | Thermo Fisher Scientific | A1049201 | |
| Chemical compound or drug | Alexa Fluor 647-Conjugated Isolectin GS-IB4 From Griffonia Simplicifolia | Thermo Fisher Scientific | I32450 | For Immunofluorescence (1:500) |
| Chemical compound or drug | cOmplete Protease Inhibitor Cocktail | Roche | 11836145001 | |
| Chemical compound or drug | Pierce ECL Western Blotting Substrate | Thermo Fisher Scientific | 32106 | |
| Chemical compound or drug | Bovine Serum Albumin lyophilized powder, essentially fatty acid free | Sigma-Aldrich | A6003 | |
| Chemical compound or drug | Fluoromount-G slide mounting medium | SouthernBiotech | 0100–01 | |

*Appendix 1 Continued on next page*

*Appendix 1 Continued*

| Reagent type (species) or resource | Designation | Source or reference | Identifiers | Additional information |
|---|---|---|---|---|
| Chemical compound or drug | Sphingosine-1-Phosphate (d18:1) | Avanti Lipids | 860492 P | |
| Chemical compound or drug | sphingosine-1-phosphate-d7 | Avanti Lipids | 860659 P-1mg | |
| Chemical compound or drug | Sphingosine (d18:1) | Avanti Lipids | 860490 P | |
| Chemical compound or drug | Sphinganine (d18:0) | Avanti Lipids | 860498 P | |
| Chemical compound or drug | Recombinant Mouse VEGF-164 (carrier-free) | Biolegend | 583102 | |
| Chemical compound or drug | Acetaminophen | Sigma-Aldrich | A7085 | |
| Chemical compound or drug | Myriocin | Cayman Chemical | 63150 | |
| Chemical compound or drug | C16 Ceramide (d18:1/16:0) | Cayman Chemical | 10681 | |
| Commercial assay or kit | DC Protein Assay Reagents | Bio-Rad | 5000116 | |
| Commercial assay or kit | AST Activity Assay Kit | Sigma-Aldrich | MAK055 | |
| Commercial assay or kit | Glutathione Assay Kit | Cayman Chemical | 703002 | |
| Commercial assay or kit | Vybrant Alexa Fluor 488 Lipid Raft Labeling Kit | Thermo Fisher Scientific | V34403 | |
| Software, algorithm | Fiji | NIH | RRID:SCR_002285 | |
| Software, algorithm | Graphpad Prism 9.0 | Graphpad Software | RRID:SCR_002798 | |
| Software, algorithm | Affinity Photo | Serif (Europe) | RRID:SCR_016951 | |
| Software, algorithm | Skyline | MacCoss Lab Software | RRID: SCR_014080 | |
| Sequence-based reagent | Sptlc1Flox For | *Alexaki et al., 2017* (PMID:28100772) | RRID:MMRRC_042294-UCD | For genotyping Sptlc1 Flox allele 5'-GGG TTC TAT GGC ACA TTT G GT AAG-3' |
| Sequence-based reagent | Sptlc1Flox Rev | *Alexaki et al., 2017* (PMID:28100772) | RRID:MMRRC_042294-UCD | For genotyping Sptlc1 Flox allele 5'-CTG TTA CTT CTT GCC AGT G GA C-3' |
| Sequence-based reagent | Sptlc1 Flox Deletion For | *Alexaki et al., 2017* (PMID:28100772) | RRID:MMRRC_042294-UCD | For genotyping Sptlc1 Flox Deletion allele 5'-CAG AGC TAA TGG AAA G GT GTC-3' |
| Sequence-based reagent | Sptlc1 Flox Deletion Rev | *Alexaki et al., 2017* (PMID:28100772) | RRID:MMRRC_042294-UCD | For genotyping Sptlc1 Flox Deletion allele 5'-CTG TTA CTT CTT GCC A GT GGA C-3' |

*Appendix 1 Continued on next page*

*Appendix 1 Continued*

| Reagent type (species) or resource | Designation | Source or reference | Identifiers | Additional information |
|---|---|---|---|---|
| Sequence-based reagent | Cdh5-Cre For | *Sörensen et al., 2009* (PMID:19144989) | RRID:MGI:3848984 | For genotyping Cdh5-Cre allele5'-TCC TGA TGG TGC CTA TCC TC-3 |
| Sequence-based reagent | Cdh5-Cre Rev | *Sörensen et al., 2009* (PMID:19144989) | RRID:MGI:3848984 | For genotyping Cdh5-Cre allele 5'-CCT GTT TTG CAC GTT CAC CG-3' |
| Sequence-based reagent | Sphk1 Flox Common For | *Pappu et al., 2007* (PMID:17363629) | RRID:MMRRC_030038-UCD | For genotyping Sphk1 Flox allele 5-GGA CCT GGC TAT GGA ACC-3' |
| Sequence-based reagent | Sphk1 Flox Rev | *Pappu et al., 2007* (PMID:17363629) | RRID:MMRRC_030038-UCD | For genotyping Sphk1 Flox allele 5-ATG TTT CTT TCG AGT GAC CC-3' |
| Sequence-based reagent | Sphk1Flox Rev Deletion | *Pappu et al., 2007* (PMID:17363629) | RRID:MMRRC_030038-UCD | For genotyping Sphk1 Flox Deletion allele 5-AAT GCC TAC TGC TTA CAA TAC-3' |
| Sequence-based reagent | Sphk2 Mutant For | *Mizugishi et al., 2005* (PMID:16314531) | RRID:IMSR_JAX:019140 | For genotyping Sphk2 allele 5'-CTC GTG CTT TAC GGT ATC GC-3' |
| Sequence-based reagent | Sphk2 Common Rev | *Mizugishi et al., 2005* (PMID:16314531) | RRID:IMSR_JAX:019140 | For genotyping Sphk2 allele 5'-CAC TGC ACC CAG TGT GAA TC-3' |
| Sequence-based reagent | Sphk2 Mutant Rev | *Mizugishi et al., 2005* (PMID:16314531) | RRID:IMSR_JAX:019140 | For genotyping Sphk2 allele 5'-TCA TCC TGC TGC CCC TTA C-3' |
| Sequence-based reagent | Apom For 1 | *Christoffersen et al., 2008* (PMID:18006500) | RRID:MGI:3772574 | For genotyping Apom allele 5'-CAC CCA GCA ACT CAT CCT TT-3' |
| Sequence-based reagent | Apom For 2 | *Christoffersen et al., 2008* (PMID:18006500) | RRID:MGI:3772574 | For genotyping Apom allele 5'-GCA GCG CAT CGC CTT CTA TC-3' |
| Sequence-based reagent | Apom Rev | *Christoffersen et al., 2008* (PMID:18006500) | RRID:MGI:3772574 | For genotyping Apom allele 5'-TCT TCC CCA CAC CCT AGC TC-3' |

