## [Editor Report]

This study clearly defines an important role for SPTLC1 in vascular endothelial cell development. The data presented herein will help promote our understanding of endothelial cell metabolism and how metabolic disorders can cause vascular abnormalities.

---

## [Decision Letter]

**Decision letter after peer review:**

Thank you for submitting your article "Endothelial-derived sphingolipids are required for vascular development and systemic lipid homeostasis" for consideration by *eLife*. Your article has been reviewed by 2 peer reviewers, and the evaluation has been overseen by Edward Morrisey as the Senior Editor. The following individual involved in review of your submission has agreed to reveal their identity: Jinlong He (Reviewer #1).

The reviewers have discussed their reviews with one another, and the Senior Editor has drafted this to help you prepare a revised submission.

Essential revisions:

As outlined by the reviewers, additional mechanistic insight into the down-stream targets of sphingolipids and how these may be altered in your animal models is required for revision. These studies could take the form of assessing candidate genes/pathways (i.e. Vegf signaling) or through a more unbiased approach i.e. transcriptome analysis of animal mutants. Additional controls and manuscript editing as raised by the reviewers is also required.

*Reviewer #1 (Recommendations for the authors):*

1. It is not very clear why the authors want to include the APAP-induced liver injury in the present study. It's like an afterthought. It's better to remove this part or change it to other angiogenesis models, such as tumor angiogenesis, angiogenesis of post-myocardial infarction, or arteriogenesis.

2. There isn't enough N number for some experiments (Figure 2I, Figure 3A, Figure 3D, Figure 4C, Figure 4D, Figure 8, and Figure 2 supplement 1). Considering the variation of retinal angiogenesis in mice, it's better to have N more than five retinas in every group.

*Reviewer #2 (Recommendations for the authors):*

Schematic illustration for the metabolisms of sphingolipids would be helpful to non-expert readers.

New hypothesis by authors would become clear when graphically abstracted. Another way is to list the previous reports comparing the results in the present study.

Please correct the misprint in Figure 1 Supplement 1 C: TBIL

In Line 191, "and" is erroneously written in "an".

In Figure 4D, "vo" is written in small letter.

Figure 2 Supplement 1 should be named as Figure 3 Supplement 1.

In line 195-197, the authors described that Cer species (C24:1, C22:0, C26:1, and C26:0) and C24:0 SM were reduced in ECKO plasma. However, these descriptions did not correctly reflect figure 5B and 5D. For example, C22:0 Cer was not changed between ctrl and ECKO.

---

## [Author Response]

Reviewer #1 (Recommendations for the authors):1. It is not very clear why the authors want to include the APAP-induced liver injury in the present study. It's like an afterthought. It's better to remove this part or change it to other angiogenesis models, such as tumor angiogenesis, angiogenesis of post-myocardial infarction, or arteriogenesis.

We appreciate the reviewer’s concern that acute liver injury model could be considered to be unrelated to angiogenesis defects shown in the retina. However, APAPinduced liver injury which is mediated in part by hepatocyte ceramide levels, is used to demonstrate that endothelial supply of SL to hepatocytes has a functional consequence in this model which occurs in response to hepatocyte stress. The other models that the reviewer suggested, i.e., tumor angiogenesis, angiogenesis post-myocardial infarction or arteriogenesis are of interest and will likely be pursued in future studies.

2. There isn't enough N number for some experiments (Figure 2I, Figure 3A, Figure 3D, Figure 4C, Figure 4D, Figure 8, and Figure 2 supplement 1). Considering the variation of retinal angiogenesis in mice, it's better to have N more than five retinas in every group.

Our N numbers are derived from 2-3 litters which contains both Cre^+^ and Cre^-^ mice. Even though Mendelian ratios are expected, in some cases skewing of various genotypes is observed. Specifically, we had the following animal numbers for each indicated experiment (Figure 2I – 4 to 7; Figure 3A – (4-5); Figure 3D – (3-7); Figure 4C (3-6); Figure 4D (3-7); Figure 8 – (3-6), and Figure 2 supplement 1 (3)). In all cases, data support the conclusions even when rigorous statistical tests were used. We do not believe that adding few more samples will change the conclusions.

Reviewer #2 (Recommendations for the authors):Schematic illustration for the metabolisms of sphingolipids would be helpful to non-expert readers.New hypothesis by authors would become clear when graphically abstracted. Another way is to list the previous reports comparing the results in the present study.

We appreciate reviewer’s suggestion and have added a graphic summary in Figure

10.

Please correct the misprint in Figure 1 Supplement 1 C: TBILIn Line 191, "and" is erroneously written in "an".In Figure 4D, "vo" is written in small letter.Figure 2 Supplement 1 should be named as Figure 3 Supplement 1.In line 195-197, the authors described that Cer species (C24:1, C22:0, C26:1, and C26:0) and C24:0 SM were reduced in ECKO plasma. However, these descriptions did not correctly reflect figure 5B and 5D. For example, C22:0 Cer was not changed between ctrl and ECKO.

We thank the reviewer for pointing these errors and have corrected them in the revised manuscript.